# Dnmt1 mediates epigenetic restriction of invasive traits in clonal crayfish

J. Jaime Diaz-Larrosa, Vitor Carneiro, Katharina Hanna, Günter Raddatz & Frank Lyko ✉

*Procambarus virginalis* (marbled crayfish) is a parthenogenetically reproducing invasive crayfish species. Its global population is monoclonal, which raises questions about the mechanisms promoting their invasiveness. Here we show that environmental changes downregulate the highly conserved *Dnmt1* DNA methyltransferase in marbled crayfish. When phenocopying this effect through a dsRNA-based in vivo knockdown, we observe enhanced invasiveness-related behavioral traits. Image cytometry and single-cell RNA sequencing reveal an expansion of mature granular immune cells and depletion of hemocyte-derived neuronal precursors, which support adult neurogenesis. Whole-genome bisulfite sequencing shows that these phenotypes coincide with a global loss of gene body DNA methylation and dysregulation of nervous and immune system genes. Additionally, we observe nucleosome destabilization to be associated with transcriptional changes after methylation loss. Taken together, our findings identify Dnmt1 as a potential canalizer of cellular and organismal phenotypes, thus providing a framework for how epigenetic mechanisms modulate invasiveness.

Biological invasions are a globally underestimated problem, with pressing ecological and economic consequences, often leading to ecosystem disruption and significant financial costs[1]. The success of invasive species is frequently attributed to their ability to rapidly adapt to new environments, despite low genetic diversity[2]. This adaptability has been linked to epigenetic mechanisms, particularly DNA methylation, which can mediate phenotypic plasticity and enable organisms to thrive in diverse environments[3,4]. In fact, DNA methylation has already been implicated in a range of traits relevant to invasiveness, including stress, temperature, salinity or pollution response, albeit with little or no mechanistic insights[5,6].

Non-indigenous crayfish species represent a well-studied paradigm for successful biological invasions[7]. Unlike other freshwater species, crayfish possess the ability to disperse over land and survive outside aquatic habitats for extended periods, which enhances their invasive potential[8]. Important examples for successful crayfish invasions include *Pacifastacus leniusculus* (signal crayfish), whose aggressiveness and adaptability has made it a successful invader in Europe, California and Japan, where it has displaced the autochthonous *Cambaroides japonicus*[9,10]. Similarly, *Procambarus clarkii* (red swamp crayfish) has been shown to feed on a wide variety of amphibian species, because it can avoid the natural defenses they have created and is a more effective predator than the indigenous species[11].

Studies of invasive crayfish species have identified behavioral traits, such as increased activity, boldness, and exploratory drive, as critical factors for successful invasion[12]. Interestingly, freshwater crayfish exhibit neurogenesis throughout their lifespan, which is driven by neuronal precursors derived from a subset of hemocytes[13]. Crayfish hemocytes are multifunctional immune cells critical for survival in pathogen-rich environments. They are comprised by three major cell types: hyaline cells (HCs), semigranular cells (SGCs), and granular cells (GCs), each with distinct functions and degrees of maturation[14–16]. Hemocyte-derived neural precursors (HDNPs) migrate to the neural stem cell niche, where they differentiate into mature neurons[17,18], thus linking the immune and nervous systems into a complex regulatory network with considerable relevance for invasive success.

Division of Epigenetics, DKFZ-ZMBH Alliance, German Cancer Research Center, Heidelberg, Germany. ✉e-mail: f.lyko@dkfz.de

Physiological adaptations that infer phenotypic flexibility have been discussed in the context of several ecological models of invasiveness. For example, the Evolution of Increased Competitive Ability (EICA) hypothesis proposes that invading species may reallocate energy and resources from immunity towards growth and reproduction to enhance competitiveness[19]. Furthermore, the General-Purpose Genotype (GPG) hypothesis suggests that clonal or genetically homogenous species can achieve ecological success through a phenotype that tolerates a wide range of ecological conditions[20]. Both frameworks emphasize physiological flexibility as a main driver of invasion success. In this context, DNA methylation has been proposed to regulate plasticity by stabilizing certain developmental trajectories and buffering environmental effects, a concept referred to as epigenetic canalization[21].

*Procambarus virginalis* (marbled crayfish) is a novel parthenogenetic freshwater crayfish species[22]. It produces clonal, all-female offspring that exhibit remarkable phenotypic plasticity despite minimal genetic variation[22,23]. Since its emergence in the mid-1990s, the marbled crayfish has spread rapidly across global freshwater ecosystems, mainly because of anthropogenic introductions[24–26]. As the species thrived in diverse environments despite the lack of genetic variation, epigenetic regulation, such as DNA methylation, was suggested to play a central role in mediating its invasive potential[4]. We have shown previously that the marbled crayfish genome is methylated and that it encodes a conserved DNA methylation toolkit consisting of a Dnmt1, Dnmt3 and Tet homolog, respectively[27]. We have also provided evidence for alterations in DNA methylation patterns in response to environmental parameters[28,29], suggesting that DNA methylation may play a role in the adaptivity of *P. virginalis*.

DNA methyltransferase 1 (Dnmt1) is an evolutionary conserved enzyme responsible for the maintenance of DNA methylation patterns. It is the main enzyme for catalyzing the covalent addition of methyl groups to cytosine bases, predominantly within CpG dinucleotides[30]. This reaction produces 5-methylcytosine (5mC), the most widely studied epigenetic mark[31]. The functional consequences of DNA methylation vary significantly depending on the genomic context. The methylation of gene regulatory elements, such as promoters and enhancers, has been linked to the regulation of gene expression by modulating the recruitment of transcription factors or other transcription modifiers[32,33]. The role of DNA methylation in other regions, such as gene bodies, is less well understood, although increasingly considered as an additional regulatory mechanism of eukaryotes[34,35].

The functional analysis of Dnmt1 was pioneered through the generation of Dnmt1 knockout mice[36], which revealed an important role of this enzyme in mammalian cell fate specification[37,38]. This includes the hematopoietic system, where Dnmt1 was shown to protect hematopoietic stem cells from lineage restriction[39]. In insects like *Bemisia tabaci* (whitefly) or *Oncopeltus fasciatus* (large milkweed bug) Dnmt1 has been suggested to play a role in gametogenesis[40,41], although these observations lack a mechanistic understanding or are not clearly related to changes in DNA methylation[42].

In a previous study, we have found that global DNA methylation was significantly lower in *P. virginalis* compared to its non-invasive parent species, *Procambarus fallax*[27]. This suggested that lower methylation levels may promote invasive traits. In this study, we first ask whether *Dnmt1* expression responds to ecological changes (temperature and water quality), reflecting the new and variable conditions potentially encountered by an invading animal. After observing a pronounced downregulation of *Dnmt1* under these conditions, we developed an RNAi-based in vivo knockdown approach in marbled crayfish to assess the functional consequences of *Dnmt1* depletion. Our results show that the reduction of *Dnmt1* promotes behavioral traits associated with successful invasions. This effect is accompanied by a shift in hemocytic cell differentiation, as knockdown of *Dnmt1*

depletes the numbers of immature HDNPs, while expanding those of the mature immune GCs. Further analysis shows a global reduction of gene body DNA methylation and transcriptional dysregulation of genes involved in nervous and immune system functions, potentially caused by a decrease in nucleosome positioning stability. Together, our findings highlight Dnmt1 as an important mediator of invasive behavior through its effects on cellular differentiation and gene expression.

## Results

### Environmental changes downregulate *Dnmt1* expression

To test whether environmental conditions can modulate the expression of the DNA methylation machinery of marbled crayfish, we quantified the levels of *Dnmt1*, *Dnmt3*, and *Tet* mRNA in hemocytes by quantitative PCR (qPCR) after changing rearing conditions in laboratory culture. In a first experiment, we transferred the animals from room temperature (RT) to a cold environment (4 °C) for 24 h and then returned them to RT for a 24 h recovery period. Compared to animals maintained at RT throughout the whole experiment, cold-treated crayfish exhibited a clear (2.1-fold) and significant ($p < 0.05$, unpaired $t$ test) downregulation of *Dnmt1* in hemocytes which became even more pronounced (11.1-fold, $p < 0.01$, unpaired $t$ test) during the recovery phase (Fig. 1a). In contrast, *Dnmt3* remained largely unchanged under both conditions (Fig. 1b), while *Tet* expression displayed a transient upregulation during the cold challenge that returned to baseline levels during recovery (Fig. 1c). Given the pronounced downregulation of *Dnmt1* following cold exposure, we next asked whether other types of environmental shifts could elicit similar effects. Therefore, we transferred animals that were reared in a biofloc environment[39] to a clearwater aquarium environment. Two days post-transfer, *Dnmt1* expression in hemocytes was strongly (4.3-fold) and significantly ($p < 0.05$, unpaired $t$ test) downregulated compared to animals exclusively reared in the aquarium (Fig. 1d). Expression levels remained lower at seven- and fourteen-days post-transfer, although these differences were less pronounced and not statistically significant (Fig. 1d). These findings strongly suggest that changes in environmental conditions downregulate *Dnmt1* mRNA expression.

### Establishment of an in vivo *Dnmt1* knockdown in marbled crayfish

To investigate the functional role of *Dnmt1* in marbled crayfish adaptability and physiology, we established an in vivo knockdown (KD) of Dnmt1. We adapted an RNA interference (RNAi) approach from other species[43] by generating ~500 bp double-stranded RNAs (dsRNAs) for both *P. virginalis Dnmt1* and *A. victoria GFP* (as a control) via in vitro transcription. After processing, the resulting dsRNAs were then injected into the crayfish hemolymph through the arthrodial membrane between the cephalothorax and the abdomen[44], which reduces the excessive bleeding of the more traditional approach through the sinus between the third and fourth walking leg[45].

To assess the efficacy of the knockdown, hemocytes were collected 48 hours post injection (hpi) using a hypodermic syringe and processed immediately for RNA isolation. qPCR showed that the expression of *Dnmt1* was robustly (63%) and significantly ($p < 0.01$, unpaired $t$ test) reduced in hemocytes from KD animals relative to controls. This depletion increased with time, resulting in a more pronounced (79%) reduction at 14 days-post injection (dpi) and 74% at 28 dpi (Fig. 1e). Analysis of *Dnmt1* mRNA levels in other tissues 14 dpi revealed different degrees of knockdown efficiency with the highest levels of reduction in heart (93%; Fig. 1f), abdominal muscle (85%; Fig. 1g) and hepatopancreas (83%; Fig. 1h). Other organs like brain, gills, ovary and the hematopoietic tissue (HPT), showed more moderate *Dnmt1* depletion (52–14%, Fig. 1i–l). These findings demonstrate that our RNAi protocol produces a stable and systemic in vivo knockdown of *Dnmt1* in marbled crayfish.

## *Dnmt1* knockdown enhances behavioral traits associated with invasiveness in marbled crayfish

To evaluate the role of Dnmt1 in marbled crayfish invasiveness, we conducted behavioral assays using a plus-shaped maze[46]. Following one week of isolation from other animals to remove any previous social memory[47], three independent batches of KD and control animals with similar weight distribution (Supplementary Fig. S1a, b) were evaluated under identical conditions. The results revealed that depletion of *Dnmt1* expression significantly reduced the total time animals remained immobile (47%, $p < 0.05$; unpaired $t$ test; Fig. 2a) and the average duration of individual stops (54%, $p = 0.06$; unpaired $t$ test; Fig. 2b), indicating increased activity levels in KD animals. Furthermore, KD animals exhibited a significant ($p < 0.05$; unpaired $t$ test; Fig. 2c) two-fold increase in the average number of climbing attempts and a 46% increase in movement frequency between maze sections (Fig. 2d), suggesting enhanced boldness and exploratory drive, which have previously been linked to crayfish invasiveness[12].

Interestingly, *Dnmt1* KD animals also presented behavioral traits indicative of reduced anxiety. They spent 31% more time in the light zones of the maze compared to controls, despite crayfish natural aversion of well-lit areas (Fig. 2e). In addition, the KD group exhibited a 46% reduction in the number of retreats (Fig. 2f), defined as incomplete attempts to enter other sections, although these differences were not statistically significant. Principal component analysis (PCA) of all the parameters measured showed a statistically significant ($p < 0.05$; Monte Carlo Permutation test) separation of the treatment groups, consistent with a global difference in behavior. Collectively, these findings demonstrate the influence of *Dnmt1* in modulating *P. virginalis* behavior and that reduced expression of Dnmt1 can result in a behavioral pattern typical for successful crayfish invaders[48].

## Knockdown of *Dnmt1* affects cell fates in hemocytic cell populations

We also examined hemocyte populations to understand the cellular basis of the observed behavioral alterations. Hemocytes are the cellular component of crayfish immunity, crucial for pathogen defense, encapsulation and surveillance[49]. The three main immune cell types develop along a single developmental trajectory from HCs to SGCs and

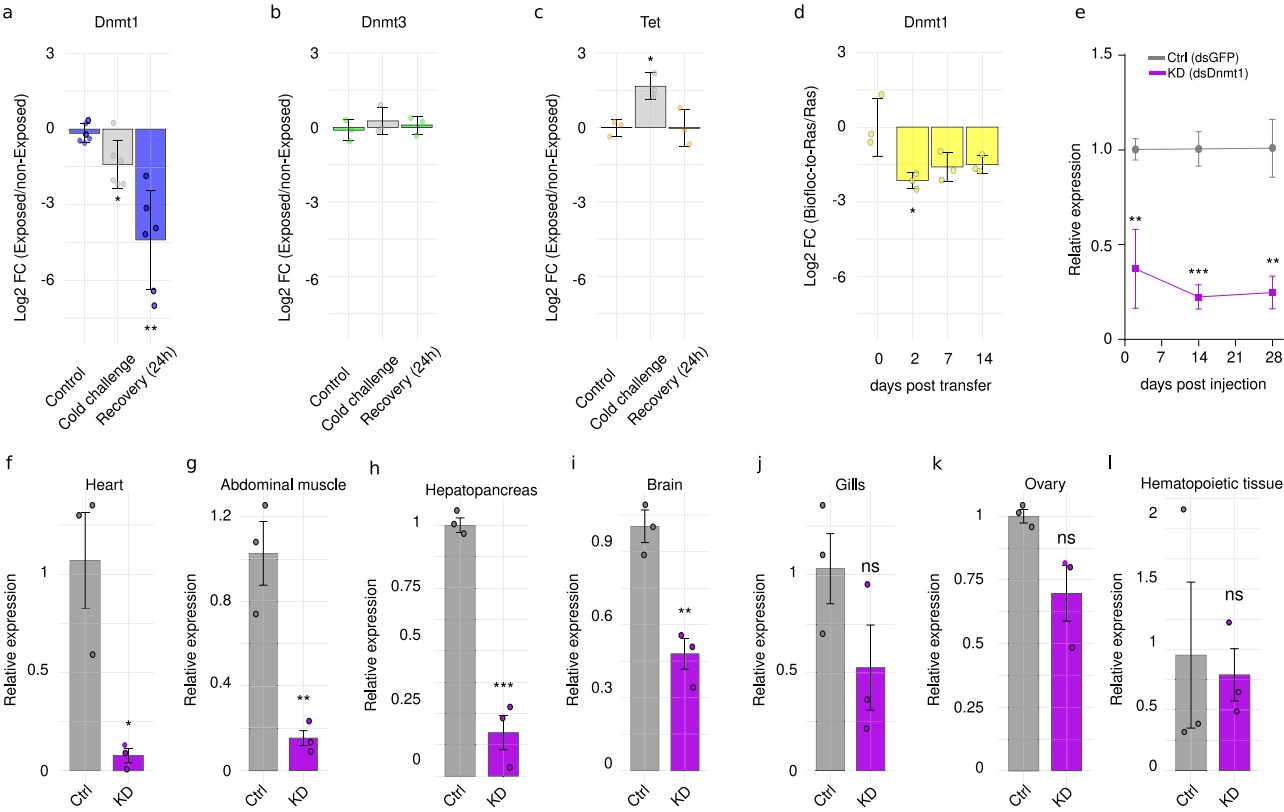

**Fig. 1 | Downregulation of *Dnmt1* in marbled crayfish through exposure to different environmental conditions and dsRNA-mediated in vivo knockdown. a** Relative *Dnmt1* expression (Log₂-fold change, FC) in hemocytes after 24 h cold exposure and 24 h at RT exposure, normalized to $t = 0$ and expressed relative to non-exposed controls; $n = 6$ biological replicates per group. Exact $p$-values: $t = 0$ vs 24 h, $p = 0.0228$; $t = 0$ vs 48 h, $p = 0.0028$; 24 h vs 48 h, $p = 0.0113$. **b** Relative *Dnmt3* expression (log₂FC) under the same cold-exposure and recovery conditions ($n = 3$ biological replicates per group). Exact $p$-values: $t = 0$ vs 24 h, $p = 0.4072$; $t = 0$ vs 48 h, $p = 0.6836$; 24 h vs 48 h, $p = 0.5616$. **c** Relative *Tet* expression (log₂FC) under the same cold-exposure and recovery conditions ($n = 3$ biological replicates per group). Exact $p$-values: $t = 0$ vs 24 h, $p = 0.0161$; $t = 0$ vs 48 h, $p = 0.0378$; 24 h vs 48 h, $p = 0.9569$. **d** Relative *Dnmt1* expression (Log₂FC) at 0, 2, 7, and 14 days posttransfer from a biofloc environment to clearwater, normalized to $t = 0$ and expressed relative to non-transferred controls; $n = 3$ biological replicates per time point. Relative *Dnmt1* expression (log₂FC) at 0, 2, 7, and 14 days following transfer

from a biofloc environment to clearwater, normalized to $t = 0$ and expressed relative to non-transferred controls ($n = 3$ biological replicates per time point). Exact p-values: $t = 0$ vs 2 d, $p = 0.0385$; $t = 0$ vs 7 d, $p = 0.1030$; $t = 0$ vs 14 d, $p = 0.1009$; 2 d vs 7 d, $p = 0.2328$; 2 d vs 14 d, $p = 0.0814$; 7 d vs 14 d, $p = 0.8228$. **e** *Dnmt1* transcript levels in hemocytes at 2, 14, and 28 days post-injection (dpi); $n = 3$ biological replicates per group and time point. Exact p-values: 2 dpi Ctrl (gray) vs KD (purple), $p = 0.0072$; 14 dpi Ctrl vs KD, $p = 0.0003$; 28 dpi Ctrl vs KD, $p = 0.0098$. **f**–**l** *Dnmt1* transcript levels in different tissues at 14 dpi ($n = 3$ biological replicates per tissue): (**f**) heart ($p = 0.0164$); (**g**) abdominal muscle ($p = 0.005$); (**h**) hepatopancreas ($p = 0.0004$); (**i**) brain ($p = 0.0047$); (**j**) gills ($p = 0.205$); (**k**) ovary ($p = 0.056$); and (**l**) hematopoietic tissue ($p = 0.81$). Each dot represents one biological replicate; bars indicate mean values ± standard deviation. Statistical analyses were performed using unpaired two-sided $t$ tests and 95% confidence intervals. ns, not significant; *$p < 0.05$; **$p < 0.01$; ***$p < 0.001$. Ctrl, control; KD, knockdown. Source data are provided as a Source Data file.

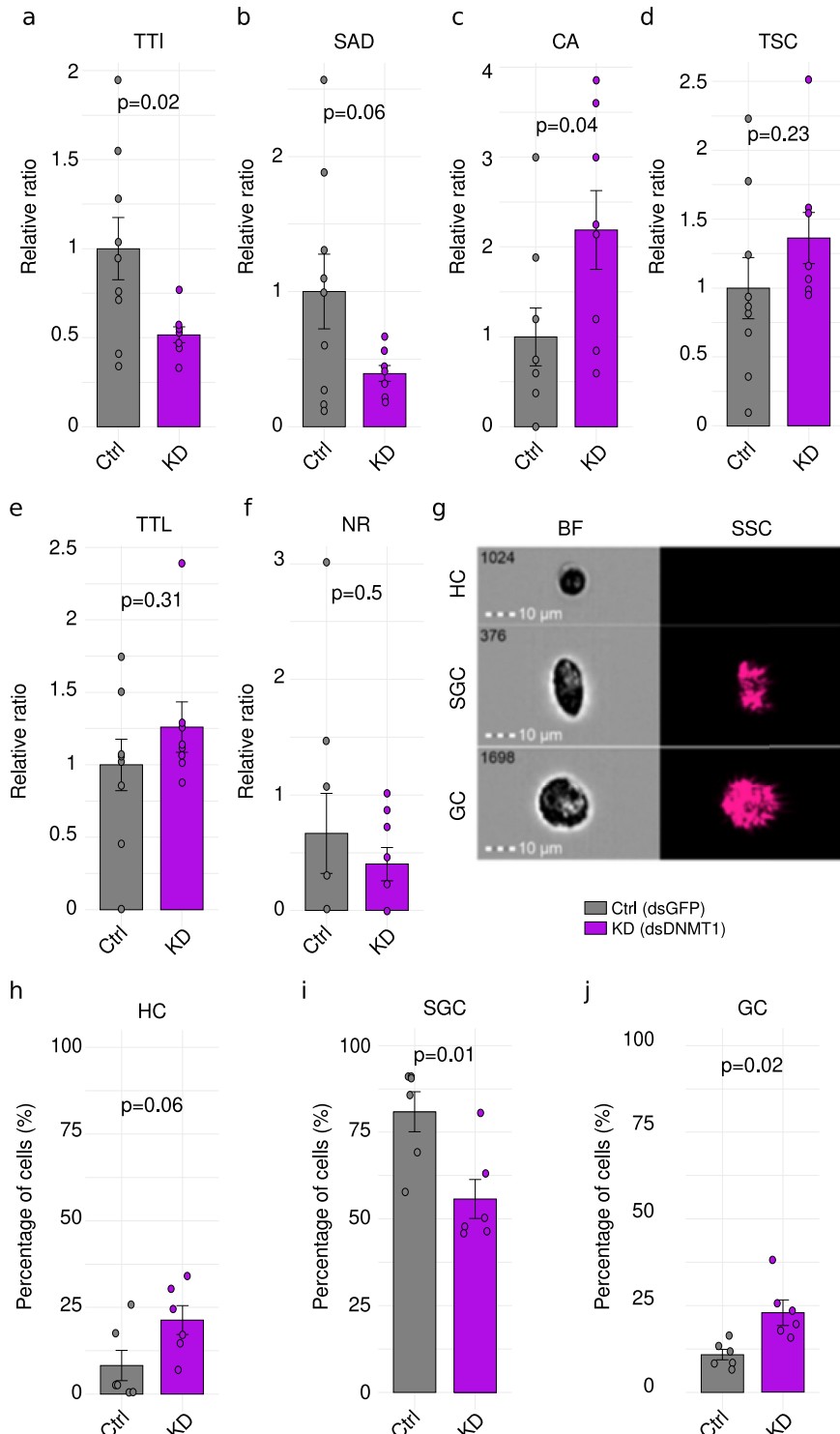

**Fig. 2 | *Dnmt1* knockdown enhances invasive phenotypes in marbled crayfish.** Quantification of behavioral traits associated with invasiveness in *Dnmt1* knockdown (KD; *n* = 8; purple) animals versus controls (Ctrl, *n* = 9; gray): **a** Total time immobile (TTI). **b** Stop average duration (SAD). **c** Climbing attempts (CA). **d** Total space changes (TSC). **e** Total time in light (TTL). **f** Number of retreats. **g** Representative image cytometry pictures showing the differences of the three main immune cell types based on brightfield (BF; size) and side scatter (SSC; granularity) parameters: HCs, SGCs and GCs. Once identified, these cells were quantified as shown in (**h**) HCs; (**i**) SGCs; and (**j**) GCs. in control (*n* = 6) and KD (*n* = 6) hemocytes at 28 days post-injection (dpi). Bars indicate mean values ± standard error. Statistical analysis used unpaired two-sided *t* tests or Welch's test depending on the equality of the variances determined using Levene's tests, and 95% confidence intervals. Source data are provided as a Source Data file.

finally to GCs[16]. In addition to their immune functions, hemocytes also serve as a source of neuronal progenitors, here termed HDNPs, which migrate to the neural stem cell niche to produce new neurons[13] (Supplementary Fig. S1c). Due to the absence of antibody-based techniques for labeling these cells, we developed an image cytometry-based protocol that enables high-resolution imaging of individual hemocytes from the three main immune cell types (Fig. 2g). This approach allowed us to classify and quantify hemocyte populations based on

morphological characteristics, such as cell size and granularity (Supplementary Fig. S1d). We analyzed one thousand hemocytes per sample from six control and six *Dnmt1* KD animals at 28 dpi. The quantification of these cell types showed that KD animals presented a significant ($p < 0.05$, unpaired *t* test) two-fold increase in GCs (Fig. 2j), with a concomitant reduction of SGCs from 80% to 55% (Fig. 2i), as well as a non-significant increase of HCs (Fig. 2h). These results demonstrate that the knockdown of *Dnmt1* affects hemocyte differentiation, shifting the balance towards more differentiated GCs.

To analyze the effects of *Dnmt1* knockdown on hemocyte cell types in more detail, we performed single-cell RNA sequencing (scRNA-seq) using the 10X Genomics platform on 3 control and 3 KD samples 28 dpi (Supplementary Table S1). Analysis of the control sample with the highest number of cells ($n = 5318$) identified four transcriptionally different clusters (Fig. 3a, b). Cluster annotation was inferred using sets of differentially expressed genes and homology-based comparisons from previously published crustacean datasets. Cluster 0 (80.5%), expressed markers consistent with SGCs, including a homolog of the crustacean hematopoietic factor[50,51], Kazal-type serine protease inhibitors[52], or Cathepsin-L, found in phagocytic cells[53,54]. Cluster 1 (10.2%) showed a strong expression of *FAT1*, a protocadherin homolog related to hemocyte-derived neuronal precursors in *P. leniusculus*[55]. The expression of other neuronal markers such as Netrin receptor *UNC-5* (ref. 56), gap-junction component Innexin 2 (ref. 57) and genes related to lipid metabolism, which is essential for the correct functioning of neurons[58], further supports this cluster as HDNPs. Cluster 2 (5.2%) was characterized by the expression of mature immune cells markers like crustins[59,60], invertebrate-type lysozyme[55], several serpin proteases[61], and a C-type lectin[62], identifying the cluster as GCs. Finally, cluster 3 (4.1%) exhibited high levels of ribosomal structural components[63], *Serp2* and *CISD3*, which are characteristic of proliferative and metabolically active cells and have previously been identified as markers of hematopoietic tissue cells in other crayfish species[55]. At the same time, this cluster also expressed glycine-rich antimicrobial peptides[64], suggesting an intermediate phenotype associated with undifferentiated circulating prohemocytes (CPHs).

In subsequent steps, we integrated this control sample with the KD sample showing the highest number of cells ($n = 2869$). Comparative analysis revealed a pronounced (92%) depletion of HDNPs in KD animals, and a 2.9-fold increase of the GC population. CPHs also became markedly (90%) reduced in the KD sample, while SGCs remained largely unchanged (Fig. 3c–e). Integration of the complete dataset ($n = 3$ for each experimental group) revealed a significant upregulation of the GC signature in KD animals, both in the GC and in the SGC clusters ($p < 2e-16$, Wilcoxon Rank-sum test; Fig. 3h). Further analysis revealed that the HDNP signature was significantly downregulated in the SGC, GC, and HDNP clusters ($p < 0.01$, Wilcoxon Rank sum test; Fig. 3i), while the SGC signature was significantly reduced in the SGC and GC clusters ($p < 0.001$, Wilcoxon Rank sum test; Fig. 3g). The CPH signature was also downregulated in the SGCs ($p < 2e-16$, Wilcoxon Rank sum test) and GCs ($p < 2e-16$, Wilcoxon Rank sum test; Fig. 3j). Finally, trajectory analysis revealed a developmental relationship between the clusters, with CPHs (undifferentiated hemocytes) developing into SGCs and then committing to either a HDNP or GC cell fate (Fig. 3f). These results suggest that Dnmt1 controls the differentiation of crayfish hemocytes by influencing the decision making from SGCs to become HDNPs and not GCs.

### *Dnmt1* knockdown reduces gene body DNA methylation
To identify potential mechanisms driving the observed cellular phenotype, we investigated the effect of *Dnmt1* knockdown on the DNA methylation level. Whole-genome bisulfite sequencing (WGBS) was performed on DNA isolated from hemocytes of two control and two KD animals (28 dpi), with an average sequencing depth of 18.2 X

(Supplementary Table S2). Data analysis showed a highly significant reduction of the average CpG methylation levels from 13.8% in control animals to 8.3% in KD samples over sixteen million mapped CpGs ($p < 2e-16$, Wilcoxon rank sum test; Fig. 4a). Methylation in non-CpG contexts (CHG, CHH, CN or CNH) remained unchanged (Supplementary Fig. S2a).

As *P. virginalis* DNA methylation is known to be targeted to gene bodies[27], we also determined average gene body methylation levels in both groups. The average results showed a pronounced and consistent reduction of gene body methylation from 27.1% in controls to 16.4% in KD samples (Fig. 4b). Further analysis also confirmed that most methylated genes were affected by methylation loss in *Dnmt1* KD hemocytes in a consistent way across the replicates (Fig. 4c). These results were validated via targeted bisulfite sequencing on selected differentially methylated regions (DMRs) using hemocyte DNA from six additional animals (three KD and three control), at 28 dpi and 6-months post-injection. This confirmed the stability of the KD effects and also validated the WGBS results at a much higher sequencing depth (Supplementary Fig. S2b–d). These findings demonstrate that the knockdown of *Dnmt1* results in a pronounced and genome-wide reduction of DNA methylation.

### Loss of DNA methylation leads to changes in hemocyte gene expression
To explore the relationship between marbled crayfish gene body methylation and gene expression, bulk RNA sequencing (RNA-seq) was performed on the same four samples (two control and two KD) that were used for WGBS (Supplementary Table. S3). Differential expression analysis identified 147 significantly upregulated and 308 downregulated genes between KD and control hemocytes 28 dpi, suggesting a role of gene body methylation in gene expression regulation in crustaceans (Fig. 4d). Validation of these findings via qPCR for 18 randomly selected genes supported the RNA-seq results (Supplementary Fig. S2e).

Interestingly, among the validated genes, a group of key crustacean hematopoietic genes[15], including prophenoloxidase (*ProPO*), mannose-binding protein (*MBP*), peroxinectin and superoxide dismutase (*SOD*), showed a general upregulation in KD samples (Fig. 4e). In addition, developmental genes like *klumpfuss* or *Notch*, which are essential for hematopoiesis in other arthropods like *Drosophila melanogaster*[65], were also upregulated. Conversely, a key crustacean cytokine, astakine 1 (*Ast1*), which was previously connected to increased neuronal incorporation in the neurogenic niche[13], was downregulated, supporting the above-mentioned results. For a more systematic analysis, we integrated the RNA-seq and WGBS datasets and identified 97 hypomethylated and downregulated genes along with 24 genes that were hypomethylated and upregulated (Fig. 4f). Functional annotation of hypomethylated and downregulated genes showed a remarkable enrichment for neuron-related processes in almost 40% of the genes, as well as roles in lipid metabolism (6.3%) and cell polarity (4.8%) which are crucial for neuronal function (Fig. 4g). A similar analysis of upregulated genes did not show a clear functional enrichment (Supplementary Fig. S2f). These results suggest that the reduction of gene body methylation in KD animals affects the expression of genes associated with neuronal functions, which could explain the observed organismal and cellular phenotypes.

### Genetic features and nucleosome positioning correlate with transcriptional effects of gene body methylation
DNA methylation within gene bodies has been associated with transcriptional activation across many species[66]. We also observed a positive correlation between gene body methylation and expression for various gene categories (Supplementary Figs. S3, S4). However, while many genes were downregulated following *Dnmt1* KD as expected, a subset of them was upregulated, suggesting more

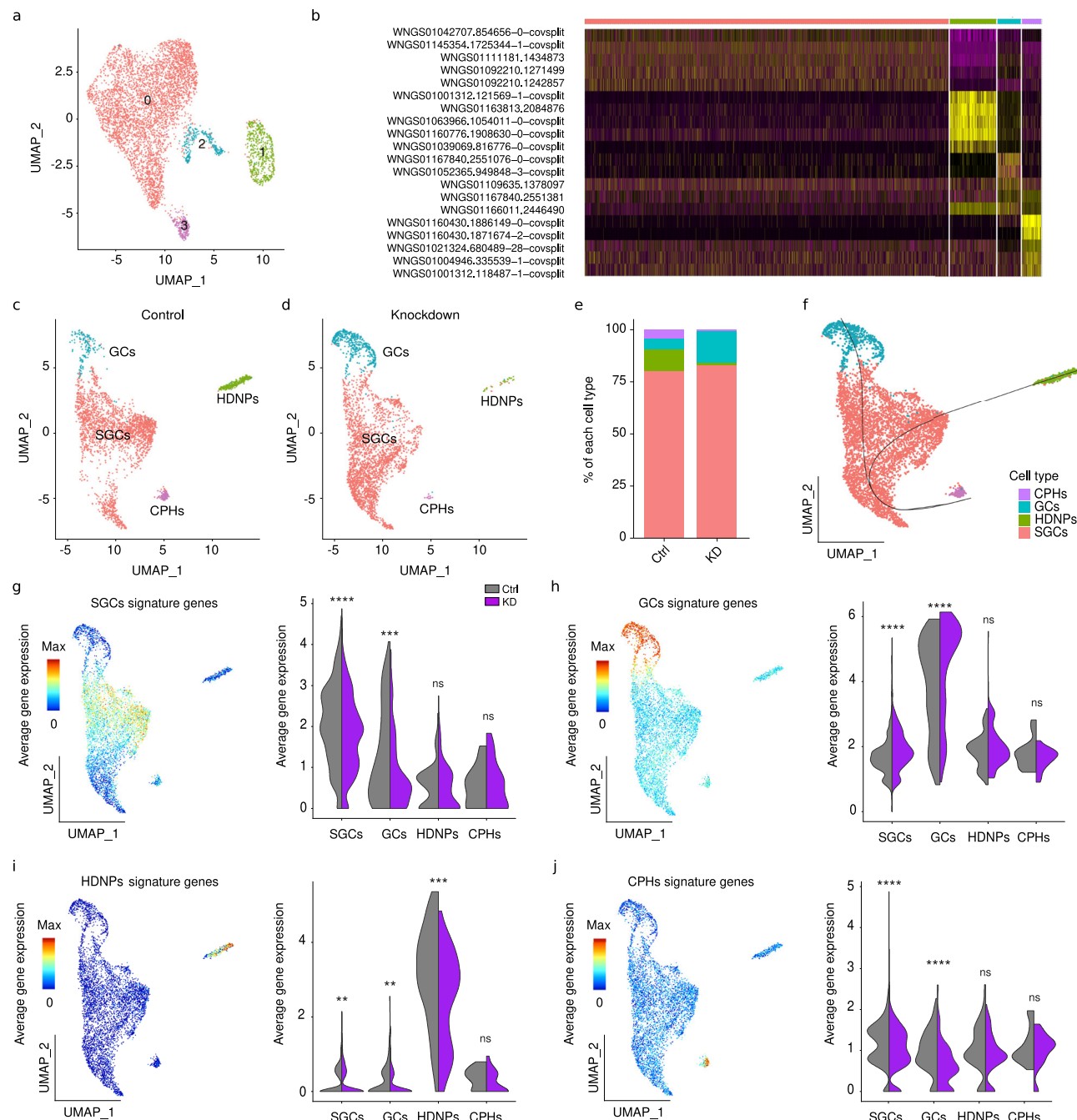

**Fig. 3 | Single-cell RNA-seq reveals a knockdown-induced reduction of putative hemocyte-derived neuronal precursors. a** UMAP projection of single-cell RNA-seq data from marbled crayfish hemocytes ($n = 5318$), colored by unsupervised Seurat clustering. **b** Identification of four transcriptionally distinct hemocyte subpopulations: semigranular cells (SGCs; cluster 0, red dots), hemocyte-derived neuronal precursors (HDNPs; cluster 1, green dots), granular cells (GCs; cluster 2, blue dots), and circulating prohemocytes (CPHs; cluster 3, purple dots). **c, d** UMAP representation of control and knockdown (KD) samples, showing marked depletion of HDNPs and expansion of GCs; CPHs are also reduced, while SGCs are largely unaffected. **e** Quantification of cell population proportions in control versus KD samples. **f** Unsupervised developmental trajectory analysis of the four major cell populations. **g–j** Expression of average key gene signatures across cell populations (combined expression shown in left heatmaps) after integration of three control (gray) and three KD (purple) samples: (**g**) SGC signature: SGCs, $p = 3.2 \times 10^{-14}$; GCs, $p = 0.0002$; HDNPs, $p = 0.94$; CPHs, $p = 1$; (**h**) GC signature: SGCs, $p < 2.2 \times 10^{-16}$; GCs, $p = 6.2 \times 10^{-8}$; HDNPs, $p = 0.47$; CPHs, $p = 0.58$; (**i**) HDNP signature: SGCs, $p = 0.006$; GCs, $p = 0.005$; HDNPs, $p = 8.72 \times 10^{-5}$; CPHs, $p = 0.89$; and (**j**) CPH signature: SGCs, $p = 4.2 \times 10^{-15}$; GCs, $p = 6.7 \times 10^{-13}$; HDNPs, $p = 0.08$; CPHs, $p = 0.28$. Violin plots show average signature expression across cell populations in control (gray) and KD (purple) samples. Statistical significance was assessed using a two-sided Wilcoxon rank-sum test and 95% confidence intervals. ns, not significant; **$p < 0.01$; ***$p < 0.001$; ****$p < 0.0001$. Source data are provided as a Source Data file.

complex underlying mechanisms. To investigate why certain genes responded differently to methylation loss, we analyzed their structural features. Our analysis revealed that hypomethylated and differentially expressed genes contained significantly more but shorter exons than genes that were unaffected (Fig. 5a, b). Furthermore, genes that were hypomethylated and downregulated exhibited significantly shorter introns and lower CpG density than upregulated genes (Fig. 5c, d). These observations suggest that gene structure influences

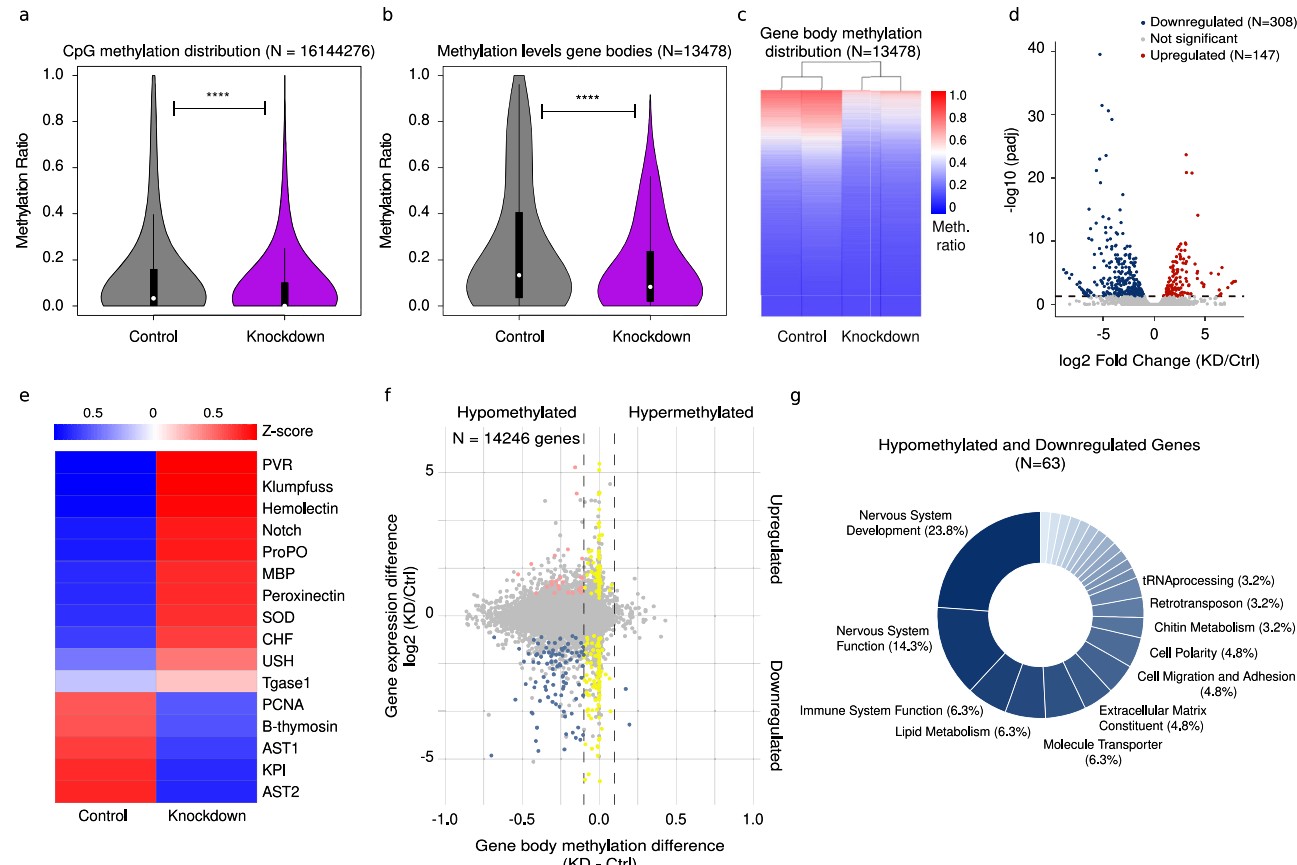

**Fig. 4 | *Dnmt1* knockdown reduces global gene body methylation and alters hemocyte gene expression in marbled crayfish. a** Average CpG methylation levels in control (*n* = 2, gray) and knockdown (*n* = 2, purple) animals. **b** Average gene body methylation levels in control (*n* = 2) and KD animals (*n* = 2). **c** Heatmap of gene body methylation across 13,478 successfully mapped genes in two biologically independent replicates for control and KD animals. **d** Volcano plot of differentially expressed genes (DEGs) identified using the default DESeq2 pipeline (negative binomial generalized linear model, two-sided Wald test). *P*-values were adjusted for multiple testing using the Benjamini-Hochberg method, and genes with adjusted *p*-value < 0.05 were considered significant. Upregulated (*n* = 147) and downregulated (*n* = 308) genes in *Dnmt1* KD versus control are highlighted. **e** Heatmap showing expression changes in known crayfish and arthropod hematopoietic genes as averaged Z-scores: *PVR* (PDGF/VEGF receptor), *ProPo* (prophenoloxidase), *MBP*

(mannose-binding protein), *SOD* (superoxide dismutase), *CHF* (crustacean hematopoietic factor), *Tgase1* (transglutaminase 1), *AST1* (astakine 1), *KPI* (Kazal protease inhibitor), and *AST2* (astakine 2). **f** Integrative methylome-transcriptome analysis identifying both differentially methylated and differentially expressed genes. **g** Doughnut plot summarizing the functional annotation of differentially methylated and downregulated genes. Categories with only one gene (1.6%) are not shown. Panels (**a**, **b**) show violin plots with embedded box plots (median, IQR, and whiskers extending to 1.5 x IQR). Statistical analyses were performed using a two-sided Wilcoxon rank-sum test and 95% confidence intervals. Exact *p*-values: (**a**) $p < 2.2 \times 10^{-16}$; (**b**) $p < 2.2 \times 10^{-16}$. Statistical analyses were performed using a two-sided Wilcoxon rank sum test; ****$p < 0.0001$. Ctrl, control; KD, knockdown. Source data are provided as a Source Data file.

transcriptional dependency on gene body methylation, with intron length playing a role in how methylation loss affects expression (Fig. 5e).

Finally, given the established link between gene body DNA methylation and nucleosome positioning[67], we examined how the loss of Dnmt1 affects chromatin organization in marbled crayfish. We therefore performed Micrococcal Nuclease sequencing (MNase-seq) on hemocytes from four control and six Dnmt1 KD animals (Supplementary Table S4). Global analysis of nucleosome positioning on hypomethylated genes revealed no significant differences in nucleosome occupancy (p > 0.05; Wilcoxon Rank Sum test; Fig. 5f) and minor changes in overall distribution (p < 0.001; Kolgomorov-Smirnov test). However, after integration of the transcriptomic data, we observed that genes which were both differentially methylated and differentially expressed upon Dnmt1 reduction (Fig. 5g and h) had a strongly different nucleosome distribution (p < 0.0001, Kolgomorov-Smirnov test), characterized by a less consistent spacing between the nucleosomes, and significantly reduced occupancy levels (p < 0.0001, Wilcoxon Rank sum test). These results suggest that loss of methylation

alone is not sufficient to alter gene expression; instead, transcriptional changes seem to require a change in nucleosomal positioning.

To further quantify this effect, we calculated the percentage of gene body regions exhibiting a substantial change in nucleosome stability or fuzziness. *Dnmt1* KD resulted in a significant reduction in the proportion of gene body regions with stably positioned nucleosomes in both differentially methylated and differentially expressed genes (p < 2e-16 in both comparisons, Wilcoxon Rank sum test; Fig. 5i, k). An increase in nucleosome fuzziness was specifically observed in differentially methylated and downregulated genes (p < 0.001, Wilcoxon Rank sum test; Fig. 5j). Together, these findings strongly suggest a link between gene body methylation, gene expression and nucleosome positioning, although the establishment of a direct causal relationship will require further experimentation.

## Discussion
Biological invasions pose major ecological and economic challenges, yet the molecular mechanisms allowing invasive species to expand remain poorly understood. While genetic factors contribute to

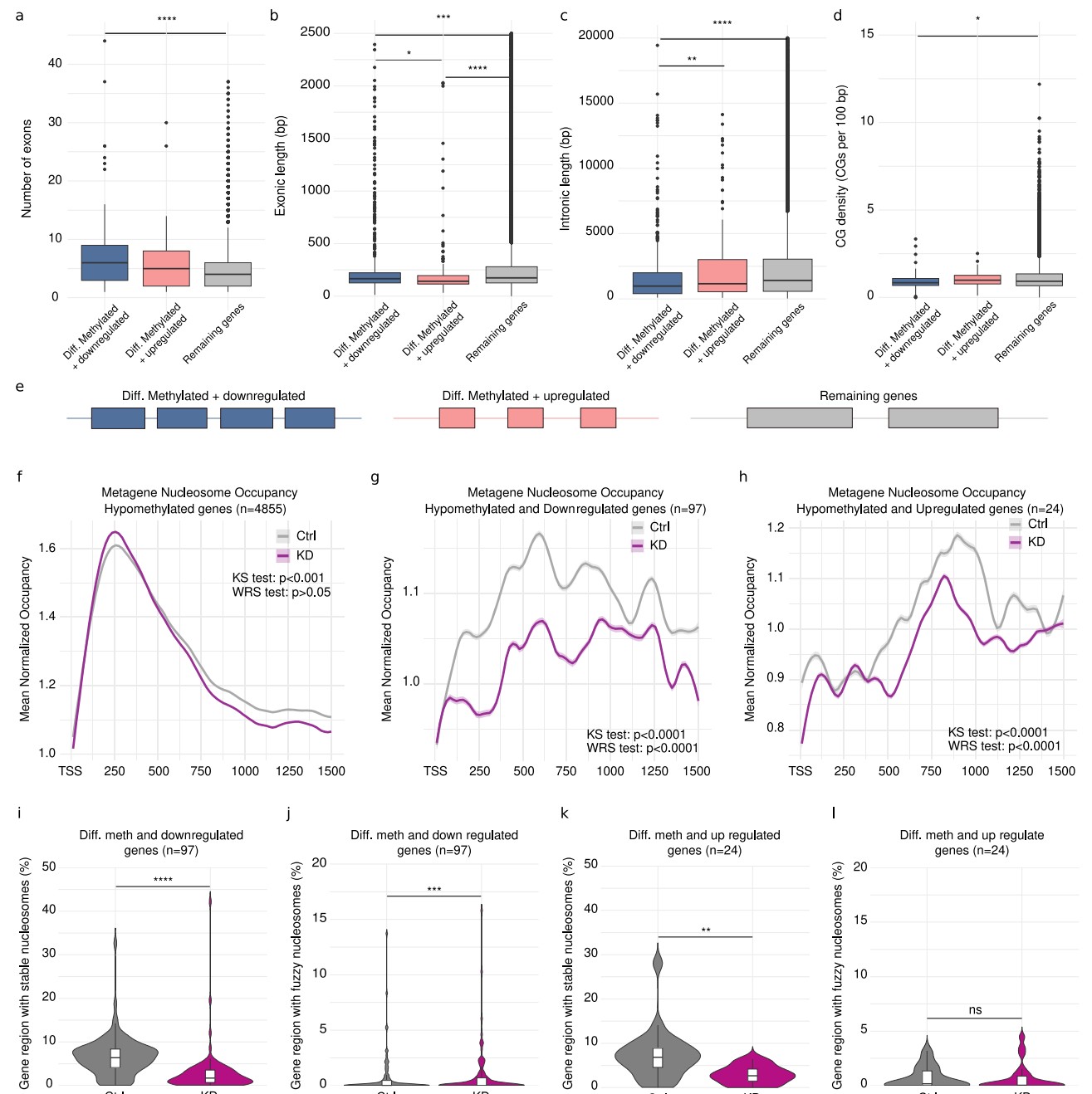

**Fig. 5 | Integrated gene structure, DNA methylation and nucleosome positioning analysis in marbled crayfish hemocytes.** Integration of matching WGBS and RNA-seq samples from two control (Ctrl) and two *Dnmt1* knockdown (KD) animals revealed that: (**a**) differentially methylated and downregulated genes (DMDR, *n* = 97) have a significantly higher number of exons than remaining genes (RG; *n* = 14126), whereas differentially methylated and upregulated genes (DMUR; *n* = 24) show a non-significant increase. **b** Exons from differentially methylated and differentially expressed genes are significantly shorter. **c** Hypomethylated and downregulated genes have significantly shorter introns. **d** Hypomethylated and downregulated genes show reduced CpG density compared to remaining genes. Statistical analyses were performed using two-sided Wilcoxon rank-sum tests (WRS). Exact *p*-values are: (**a**) DMDR vs RG, *p* = 8.1 x 10⁻⁵; DMUR vs RG, *p* = 0.12; DMDR vs DMUR, *p* = 0.66. **b** DMDR vs RG, *p* = 5.8 x 10⁻⁴; DMUR vs RG, *p* = 2.9 x 10⁻⁵; DMDR vs DMUR, *p* < 2.2 × 10⁻¹⁶; DMUR vs RG, *p* = 0.2; DMDR vs DMUR, *p* = 0.007. **d** DMDR vs RG, *p* = 0.03; DMUR vs RG, *p* = 0.77; DMDR vs DMUR, *p* = 0.15. **e** Schematic summarizing the gene structure differences described in panels (**a**–**d**). **f** Metagene nucleosome occupancy profiles of hypomethylated genes (*n* = 4855) over the first 1500 bp downstream of the transcription start site

(TSS) show a discrete change in profile distribution upon *Dnmt1* knockdown. **g, h** Similar analyses of differentially methylated and (**g**) downregulated; and (**h**) upregulated genes reveal a more pronounced disturbance in nucleosome positioning, periodicity, and occupancy. All metagene plots show mean occupancy profiles ± standard error (shaded). Differences in nucleosome occupancy profiles between Ctrl (gray) and KD (purple) were evaluated using two-sided WRS tests, and differences in distribution using two-sided Kolmogorov-Smirnov (KS) tests. Exact *p*-values are: (**f**) KS *p* = 1.3 × 10⁻⁴, WRS *p* = 0.053; (**g**) KS *p* < 2.2 × 10⁻¹⁶, WRS *p* = 3.2 × 10⁻⁹; (**h**) KS *p* = 1.03 × 10⁻¹¹, WRS *p* = 3.1 × 10⁻¹⁰. **i–l** Quantification of the percentage of gene regions with stable and fuzzy nucleosomes in differentially methylated and differentially expressed genes. Statistical analyses were performed using two-sided WRS. Exact *p*-values for stable and fuzzy nucleosome coverage are: (**i**) *p* = 9.7 × 10⁻¹³; (**j**) *p* = 6.5 × 10⁻⁴; (**k**) *p* = 0.003; (**l**) *p* = 0.8. Confidence intervals are always 95%, and box plots in this figure indicate the median (horizontal line), interquartile range (25ᵗʰ-75ᵗʰ percentile), and whiskers extending to 1.5 x the interquartile range. ns, not significant; *p < 0.05; **p < 0.01; ***p < 0.001; ****p < 0.0001. Source data are provided as a Source Data file.

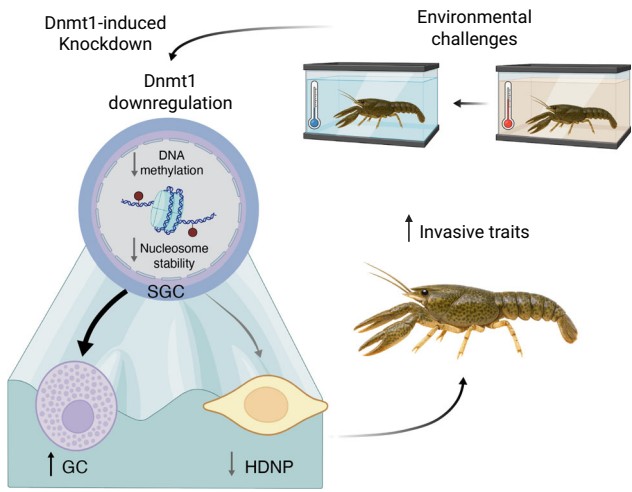

**Fig. 6 | Model of *Dnmt1*-mediated gene body methylation in regulating nucleosome stability, gene expression, hemocyte development, and invasive behaviour in marbled crayfish.** Environmental perturbations, such as acute temperature shifts or exposure to novel habitats, lead to a downregulation of *Dnmt1*, which can be recapitulated via dsRNA-mediated knockdown of *Dnmt1*. This results in a global loss of gene body DNA methylation, which destabilizes nucleosome positioning, particularly within specific gene classes. The disruption of chromatin architecture alters transcription, leading to DEGs associated with immune and neuronal functions. These molecular changes shift the normal differentiation trajectory of hemocytes. The system becomes biased toward the production of GCs, the most mature immune cell type, at the expense of HDNPs, which are required for adult neurogenesis. These changes culminate in measurable organismal phenotypes, including a potentially enhanced immune system and increased boldness, activity, and exploratory behavior, commonly associated with successful biological invasions. This model positions Dnmt1 as a key molecular integrator of environmental cues and a central regulator of phenotypic plasticity in a clonal species. Created in BioRender. Diaz, J. (https://BioRender.com/7tx3dzv) is licensed under CC BY 4.0.

adaptability over the long term, epigenetic modifications such as DNA methylation can modulate phenotypic plasticity without modifying the DNA sequence, thus allowing species to rapidly adapt to new environments[4,68]. Here, we demonstrate that Dnmt1-mediated gene body methylation regulates behavioral traits associated with successful invasions in the clonal invasive marbled crayfish. Our discoveries establish a mechanistic link between epigenetics, canalization, and ecological adaptability, highlighting the role of DNA methylation in invasiveness (Fig. 6).

DNA methylation is well known for its role in behavioral modulation, especially in mammals, where it influences stress response and psychiatric disorders[69,70]. In addition, it has been shown in *Apis mellifera* (honeybees) that task specialization within worker subcastes is associated with differential methylation of gene bodies, some of which were related to cognition[71]. Here we show that in vivo *Dnmt1* knockdown enhances behavioral traits associated with freshwater crayfish invasiveness, such as increased activity, boldness, and exploration drive[12]. These behavioral changes coincide with a reduction of putative HDNPs and a consistent downregulation of genes associated with neuronal physiology and development. Mechanistically, these changes were accompanied by a reduction and destabilization of nucleosome positioning, which became more fuzzy upon loss of gene body methylation in differentially expressed genes. Nucleosome stability has been implicated in transcriptional regulation, and a correct positioning and spacing is potentially necessary to activate or inhibit transcription start and elongation[72,73]. Interestingly, nucleosome positioning stability is dependent on gene length, and less conserved in regions that are distant from the transcription start site[74]. Consistent with this, we observed a pronounced transcriptional deregulation of

short genes in the absence of Dnmt1, showing an increased sensibility to the loss of DNA methylation.

*Dnmt1* KD in marbled crayfish also affected cellular differentiation in hemocytes, the main immune cells in decapod crustaceans[14]. We observed a skewed differentiation towards GCs, the most mature immune cell type, reducing the numbers of HDNPs. This suggests that gene body methylation contributes to cell-fate specification in marbled crayfish. Interestingly, a similar methylation-dependent lineage restriction has long been known in mammals, where Dnmt1-deficient HSCs lose lymphoid potential in favor of myeloid differentiation[39]. Similarly, lack of Dnmt1 in mouse neural progenitors biases differentiation towards astrocytes at the expense of neurons[75]. These findings are consistent with Waddington's concept of canalization, in which epigenetic mechanisms lead to developmental equilibrium or *homeorhesis* by stabilizing certain pathways or *chreods* to reach defined end-states[21,76]. Our findings suggest that Dnmt1-dependent gene body DNA methylation is one of such mechanisms in invertebrates.

The relationship between immunity and invasive success has long been debated in the ecoimmunology field. The refined Evolution of Increased Competitive Ability (EICA) hypothesis[19] suggests that a dampened immune system favors invasion by decreasing the energetic costs of inflammatory responses and reallocating resources to growth and reproduction. However, the experimental confirmation of the EICA-refined hypothesis has remained inconsistent. In fact, studies comparing *Drosophila suzuki* to *D. melanogaster*[77] and *Harmonia axyridis* to *Tribolium casteneum*[78] suggest that successful invasive species present higher hemocytic counts and diversity of antimicrobial peptides, forming a more robust immune system. This notion is also supported by other studies in vertebrates[79]. The increase of GC counts and the upregulation of effector genes in our *Dnmt1* KD crayfish also suggest a potentially stronger immune system, which could be advantageous in new environments.

*Procambarus virginalis* has established stable populations in over twenty countries and across diverse biotopes[24,25]. Interestingly, clonal reproduction is highly overrepresented among invasive species[2]. In fact, 70% of the plants reported in the IUCN Global Invasive Species Database are clonal[2]. Clonality is also seen in invasive animals, such as the queenless *Ooceraea biroi* (clonal raider ant), whose workers reproduce via automictic parthenogenesis, allowing the species to expand across diverse countries[80]. Epigenetic modulation has been proposed as an attractive mechanism to enable the General-Purpose Genotype[20] that defines clonal invasive species. As we found that lower DNA methylation enhances behavioral traits associated with invasiveness, our findings suggest that epigenetic relaxation (through hypomethylation) decreases canalization, thereby promoting invasiveness in *P. virginalis*. These findings are consistent with the idea that decreased global DNA methylation is as a signature of early phases of invasions[3]. Aquatic invertebrates recently introduced into new environments like *Ficopomatus enigmaticus* (tubeworm), *Xenostrobus secures* (pygmy mussel), and the ascidian *Didemnum vexillum* (sea vomit), have all shown evidence for DNA hypomethylation when compared to older or non-invasive populations[6,81,82]. Similarly, we have observed lower methylation levels in marbled crayfish compared to its parent species, *P. fallax*[27].

Altogether, our study suggests a multilayered role for DNA methylation in the physiology of marbled crayfish, where *Dnmt1* KD and loss of gene body DNA methylation destabilize the normal nucleosomal architecture, affecting transcription. This dysregulation disrupts the normal process of hemocyte differentiation, enhancing invasive behavior, contributing to the ecological adaptability of an invasive species. While our research identifies Dnmt1 as an important regulator of marbled crayfish physiology and invasion, the demonstration of a direct causal relationship between methylation, expression and nucleosome distribution will require additional experiments,

such as time-resolved analyses or the functional rescue of Dnmt1 activity. In addition, the upstream mechanisms that control Dnmt1 activity and expression remain elusive. In mammals, post-translational modifications, such as phosphorylation by the AKT1 kinase, have been shown to inhibit Dnmt1 degradation[30,83]. As this kinase is part of the mTOR pathway and becomes activated in the presence of nutrients[84], lack of resources, as possibly encountered during the early phases of an invasion, before an ecological niche is acquired, could lead to Dnmt1 degradation and hypomethylation. Finally, given that *Dnmt1* knockdown impairs HDNPs development, further research should explore how reduced Dnmt1 activity affects adult neurogenesis by limiting the replenishment of neurons from hemocytic progenitors. A decrease in this neuronal renewal mainly affects clusters innervating the olfactory and accessory lobes, which are essential for detecting and integrating olfactory, visual and mechanosensory signals[17]. As such, this effect may alter sensory processing, stress responsiveness and overall behavior, impacting adaptation to novel environments.

Finally, while our study does not directly address ecological management, identifying molecular mechanisms mediating phenotypic plasticity and rapid adaptation could inform early detection or mitigation strategies. For instance, recognizing epigenetic relaxation (e.g., global hypomethylation) as a potential biomarker of invasiveness might help prioritize populations for monitoring, thereby contributing to ecosystem protection.

## Methods

Abbreviations used throughout the manuscript are listed in Supplementary Note 1.

### Ethics approval statement
All laboratory experiments were conducted by approval of the institutional animal welfare committee, in compliance with regional standards guidelines, and regularly monitored by the authorities (Permits 35-9185.82/A-5/20 and 35-9185.82/A-7/25).

### Animal husbandry
Laboratory animals were individually kept in 26 x 18 x 14 cm plastic containers with size-appropriate shelters and environmental enrichment objects, and fed with aquarium feed three times a week. Animals were daily monitored throughout and after the experiments for indications of distress. Tap water at RT was used as the water source and replaced once a week.

### Environmental exposure experiments
To assess the impact of sudden cold stress on *Dnmt1* expression, marbled crayfish were exposed to a temperature fluctuation experiment. Three to six individuals were transferred from their standard rearing conditions at RT (20–22 °C) to a cold environment (4 °C) for 24 h. Crayfish were then returned to RT for an additional 24 h period. Hemolymph samples were collected at three time points: immediately before cold exposure ($t = 0$), after 24 h at 4 °C ($t = 1$), and after the subsequent 24 h recovery at RT ($t = 2$). Control animals remained at RT throughout the experiment and were sampled in the same way.

In a similar manner, to evaluate the effects of environmental changes associated with different aquaculture practices on *Dnmt1* expression, three individual crayfish raised in a biofloc-based system were transferred to an aquarium system and maintained for 14 days. Hemolymph samples were collected at four time points: immediately before transfer ($t = 0$), and at 2, 7, and 14 days post-transfer. Control animals were continuously housed in the aquarium system and sampled at the same time points.

### Sample collection
Hemocytes were collected following available protocols[85]. Briefly, crayfish were placed on a paper tissue with their eyes covered and their claws and pereopods immobilized. Then, the abdomen was stretched and wiped with 70% ethanol and a 20GX2" needle (0.9 X 50 mm) was introduced at the right side of one of the superior abdominal segments. Hemolymph was collected in a tube containing 0.4 ml of filtered anticoagulant solution [NaCl 0.14 M, Glucose 0.1 M, sodium citrate dihydrate 30 mM, citric acid 26 mM, EDTA 0.01 mM at pH 4.6], and mixed by inverting the tube. After centrifugation, the pellet was snap-frozen in liquid nitrogen before further use. For the collection of abdominal muscle, heart, gills, hepatopancreas, ovary, hematopoietic tissue (HPT) and brain, the animals were anesthetized in iced water for > 5 min and euthanized by decapitation. Each tissue was rinsed with PBS, snap-frozen in liquid nitrogen and stored at − 80 °C before further use. The numbers of replicates per treatment group in each experiment are available in Supplementary Table S5.

### Gene expression analysis using qPCR
Hemocytic RNA for qPCR and RNA-seq was isolated using the Ribo-Pure kit (Invitrogen, Cat. No. AM1924) according to the manufacturer's instructions. cDNA was produced using the Quantitect Reverse Transcriptase kit (Qiagen, Cat. No. 205311). qPCR analyses were performed on a LightCycler 480 Real-Time PCR System (Roche) using the MESA Green qPCR Mastermix Plus for SYBR Assay (Eurogentec, Cat. No. RT-SY2X-03 + WOU). The expression levels of the different genes were determined by the average cycle threshold (Ct) value of three technical replicates using TATA-box-binding protein (TBP) as a reference gene for normalization by the $2^{-\Delta\Delta CT}$ method[86]. Primer sequences for qPCR were generated using Primer3 and are provided in Supplementary Table S6.

### dsRNAi-based knockdown
Double-strand RNA was produced by in vitro transcription using the MEGAscript kit (Invitrogen, Cat. No. AM1334). Two oligonucleotides previously designed with eRNAi[87] were used to amplify a ~ 0.5 kb long region of the target gene by PCR from a pool of marbled crayfish cDNA generated with the Quantitect Reverse Transcriptase Kit (Qiagen, Cat. No. 205311). Subsequently, the in vitro transcribed dsRNA was column-purified following the instructions of the MEGAclear kit. dsRNA targeting *Aequorea victoria* GFP was produced as a negative control from a commercial plasmid containing that gene (LentiCRISPRv2GFP, Addgene, Cat. No. 82416). Primer sequences were generated using Primer3 and are available in Supplementary Table S7.

Animals were injected in two consecutive days with 2 µg of dsRNA per g of body weight per injection using a 30 G x 8 mm U-100 insulin syringe (BD Micro-Fine, Cat. No. 324825) through the arthrodial membrane between the cephalothorax and abdomen, while their head was covered to reduce stress[43].

### Behavioral experiments
Behavioral experiments were conducted in a subaquatic dark/light plus-shape maze[46], to measure the crayfish spontaneous exploratory behavior. Briefly, using a 50 x 50 cm arena comprising two dark (red-light intensity: 10 lux) and illuminated (white-light intensity: 70 lux) perpendicular sections with 15 cm in length and 12.5 cm in width, 15 min videos of each animal were blindly recorded using a high-definition video camera (model HC-380, Panasonic). Videos were analyzed using BORIS software (v.8.27.10), which allowed for the tracking of different parameters. Parameters measured included anxiety[46] and invasiveness traits[12], i.e., total time in light (TTL; seconds in light arms / total seconds of recording), retreat ratio (NR; number of retreats / total number of change of space), total immobile time (TTI; seconds the animal spends immobile / total seconds of recording), stop average duration (SAD; total time duration of stops / number of times the animal stops), change of spaces (TSC; number of times the animal moves from light to darkness or vice versa), and climbing attempts (CA; number of times the animal goes against a wall of the

maze and incorporates its body 45°–90°). Animals were isolated in opaque tanks to remove any effect of previously established social memory one week before the recordings. Three independent batches of similar weight animals (Supplementary Fig. S1), three control and three KD animals per batch (total $n = 9$ Ctrl and 9 KD animals per group) were recorded. One KD animal, which presented no locomotion, was excluded from the analysis, following established exclusion criteria[46]. Each parameter was normalized to the control biological replicates in each batch and Levene's statistical tests (from R package *car*) were performed to measure the equality of the variances. Whenever a parameter's variances were significantly unequal, a Welch's *t* test was used to compare control and knockdown performance. If Levene's test was not significant, a Student's *t* test was used. Monte Carlo Permutation tests ($n = 500$ permutations) were performed to analyze significance in the PCA using the *vegan* package in R.

## Image Cytometry

Total filtered hemolymph was pelleted and resuspended in 1% dilution of paraformaldehyde in water, where it was fixed for 20 min at RT. Cell-type proportions were assessed using an Image Cytometer ImageStream X MKII from Merck Millipore. One thousand cell events were acquired from each sample using the INSPIRE software. Fluidics speed was adjusted to *Low,* and magnification at 40X. Channels used included the standard bright field configuration in channels 01 and 09, which allowed for cell visualization and side scatter detection in channel 06, ($\lambda = 745 - 785$ nm) for granularity assessment. Gating between the different cell populations was done with the IDEAS software. Briefly, the initially recorded sample population was filtered to keep the focused signals (Gradient RMS M01 Ch01 > 30) and single cells (Area M01 vs Aspect Ratio Intensity M01 Ch01). Cells were then split into smooth and rough cells based on the (Intensity MC Ch06, > 1.5e5 associated to rough cells). Using differential size and granularity (Intensity MC Ch06 vs Area Ch01) granular cells were identified as cells with an area > 200 $\mu m^2$ and a high scatter intensity ( > 1e5) and making use of the naturally observed separation between cell populations. All cells were visually inspected. *Spot Count* function which allows to count the number of spots, in our case, granules in the side scatter channel (Ch06) was used to identify the HCs as those containing 0 or 1 granules. Those cells that were not classified as either GCs or HCs were considered SGCs. Signals associated with non-cellular identities, i.e., debris, were discarded. Statistical analysis was performed using GraphPad Prism.

## scRNA sequencing

Fresh hemocytes were extracted from three control and three *Dnmt1* knockdown animals as stated above. Subsequently, about 10,000 single cells per sample were used to prepare sequencing libraries with the Chromium Single Cell Reagent kit (10X Genomics, v3.1 chemistry; Cat. No. 1000128), according to the manufacturer's protocol. Library concentration was determined with the Qubit dsDNA HS Assay kit (Life Technologies, Cat. No. Q32851), and cDNA integrity was measured with 2200 TapeStation (Agilent Technologies, Cat: No. 5067-5582). Finally, pair-end sequencing was performed with the NovaSeq 6000 device (Illumina).

## scRNA-seq data analysis

Raw sequencing reads from all samples were processed with the Cell-Ranger v7.0.0 software (10X Genomics) without forcing any cell number and including intronic counts (command cellranger count --include-introns true --transcriptome = [genome index] --fastqs= [FASTQ directory] --sample = [sample name]) and analyzed using the Seurat v4.3.0 and SeuratObject v4.1.3 R packages. Seurat objects were initialized with a minimum of 3 cells and 100 features per cell. If not specified otherwise, default parameters were used for all functions. Cells with more than 5% of mitochondrial counts as well as over 2000 or less than 20

unique feature counts were filtered out. For the integration of the six samples, down-sampling to 622 cells was performed. Data were then normalized with a scale factor of 10,000 and highly variable features were identified with the vst method, selecting the top 2000 features for downstream analysis. Data scaling was performed with 50 principal components, and clustering dimensions were adjusted according to dataset size: 9 dimensions for the single control sample with the highest number of cells, 15 for the integrated control versus KD dataset, and 11 for the full final integration. Downstream low resolutions (0.1, 0.09 and 0.1, respectively) were used throughout the analysis. Clustering analyses were repeated multiple times using identical parameters to assess robustness to stochastic variation, yielding consistent results across runs. No explicit random seed was set. The most DEGs between the detected clusters were identified using the FindAllMarkers function, considering genes present in a minimum fraction of 25% of the cells. These genes were then used for cluster association based on markers and homology to similarly annotated cell types from related crustaceans and arthropods, and the top 5 genes of each identified cluster were considered as the respective gene signature. Because validated antibody markers are not available for *P. virginalis*, annotations should be regarded as putative rather than definitive. Statistical analysis was performed using the Wilcoxon rank sum test on the six-sample integrated object.

## Trajectory inference

Slingshot (v2.6.0) was used to perform pseudotime analysis to infer lineage relationships among the characterized cell clusters. The integrated Seurat object of the highest number of cells (Ctrl1 and KD3) containing the UMAP embeddings and correct cluster labels was first processed using the SingleCellExperiment function. Only unsupervised analysis and curve lines were used for this experiment.

## DNA methylation analysis (WGBS and MiSeq)

Genomic DNA (gDNA) for WGBS and MiSeq was isolated from the respective tissue samples as described in other studies[28]. Briefly, after homogenizing the tissue, either by pipetting for hemocytes and the HPT, or by use of a Tissue Ruptor (Qiagen) for the rest. Samples were digested using proteinase K (Invitrogen, AM2546) and RNAse A (Sigma-Aldrich, R6513). gDNA was precipitated using isopropanol and its quality assessed on a 2200 TapeStation (Agilent). The EZ DNA Methylation-Gold kit (Zymo Research, Cat. No. D5006) was used for bisulfite conversion. Library amplification was performed using KAPA HIFI HotStart ReadyMix (Roche, Cat. No. 07958935001). Samples were sequenced on an Illumina HiSeq X platform using a 150 bp paired-reads protocol (PE150). Raw reads were trimmed using Trim Galore (v0.6.6) with default Phred33 and a minimum read length of 36 bp. 10 bases were clipped from the 5′ ends of both forward and reverse reads. Trimmed reads were then aligned to the *P. virginalis* reference genome using Bismark (v0.20.0) with Bowtie2 (v2.4.2) as the underlying aligner. Mapping was performed in paired-end mode with the following parameters: --score_min L,0, -0.4. Duplicate reads were then removed using the deduplicate_bismark function, and methylation calls were extracted using bismark_methylation_extractor with the options --paired-end, --bedGraph, --comprehensive and --scaffolds to obtain CpG methylation profiles in BEDGraph format. Data were processed adapting a previously published pipeline[27] filtering out those cytosines with < 10X combined coverage in all samples, and keeping annotated genes with at least 10 CpGs. Mapped cytosines were uniquely annotated to display methylation values using tabix and annotated to all transcript variants regardless of the overlapping using the intersect argument of bedtools (v2.24.0) for integration purposes. Genes were considered to have differential methylation whenever the difference between control and knockdown averages was more than | 0.1| as performed in previous studies[27]. Plots such as box plots, violin plots, heatmaps and doughnut plots were created with the specific

packages such as ggplot2 (v3.4.4), vioplot (v0.4.0) and heatmap.2. After integration with the RNA-seq data, functional annotation of the differentially methylated and differentially expressed genes was retrieved. For those genes with no annotation, a BLAST analysis of the unknown protein sequences was performed against the *D. melanogaster* proteome available in Uniprot as *UP000000803* on the 28th of February of 2024. If genes had still no ID associated, they were not taken into consideration for further analysis. For the integration with the WGBS, after transcripts per million (TPM) calculation, genes with 0 TPM were excluded, correlation level and statistical significance were calculated using the test for association/correlation *cor.test* from R. Gene classification was done after BLASTp (e-value $10^{-10}$) of crayfish proteome to sequences of *D. melanogaster* available in FlyBase for TFs, meiosis and hematopoiesis genes; a list of human housekeeping genes[88]; a list of crayfish immune genes[89]; and a list of nervous system genes from *Homarus americanus* (American lobster)[90].

For targeted bisulfite-sequencing, primers were designed to amplify the top differentially methylated regions (DMRs) identified using a sliding-window approach (1000 bp) and a Fisher's exact test performed between the four samples analyzed. Regions containing at least 15 CpGs were selected, and primer pairs were generated with the Free Bisulfite Primer Calculator (Zymo Research). All primer sequences are listed in Supplementary Table S8. Briefly, purified DNA was bisulfite converted using the EZ DNA Methylation-Gold kit and 250 ng of genomic DNA as input. Sequencing libraries were prepared and amplified using the Nextera XT Index kit v2 (Illumina, Cat. No. 15052163) and the KAPA HIFI HotStart ReadyMix. Libraries were purified using AMPure XP magnetic beads (Beckman Colter Cat. No. A63881), pooled and sequenced using the Illumina MiSeq V2 platform and paired-end 150 bp reads (PE 150 Nano). Results were analyzed using BisAMP[91] setting parameters Ccut=301 and alpha=0. Unconversion ratios per DMR were calculated by averaging the ratio of each CpG, and significance was calculated using ANOVA. Box plots were generated using gplots (v3.1.3.1).

## Bulk transcriptomics

RNA was processed using the RNA clean-up and concentrator kit (Zymo Research, Cat. No. R1013). Four samples (two control and two knockdown) were sequenced using the Illumina HiSeq 4k platform and 50 bp single reads (SR50). Raw data were mapped using Hisat2 (v2.2.1) with parameters -p 16 -q --rna-strandness R followed by sorting and indexing with Samtools (v1.9) after filtering out the second and third exon of the *Dnmt1* gene, since its sequence overlapped with the injected dsRNA-Dnmt1 and was also sequenced. Counts were produced using featureCounts (v1.5.1) with the options -S 2 -a. Genes with low counts ( < 10 across samples) were filtered out, and differential expression analysis was performed using DESeq2. TPM values were calculated and used for comparison to qPCR data and integration analysis without filtering out genes with low counts. Integration analysis with WGBS to identify groups of genes differentially methylated and expressed, was done after removing *Dnmt1* from the list of downregulated genes to avoid cofounding effects.

## Nucleosome positioning analysis

Hemocyte samples from four Ctrl and six *Dnmt1* KD animals 28 dpi were isolated and treated as described[92]. One million cells were fixed with 1% PFA for 10 min at RT, using 26 µl of 2.5 mM glycine to stop the reaction. Fixed cells were then washed with PBS and, after counting, resuspended in 300 µl swelling buffer [HEPES 25 mM, MgCl$_2$ 1 mM, KCl 10 mM, and 0.1% NP-40 at pH 7.8 and enriched with cOmplete protease inhibitor (Roche, Cat No. 11697498001)] and incubated for 10 min on ice. After centrifugation (5 min, 845 g), the pellet was resuspended in 90 µL MNase buffer (KCl 25 mM; MgCl$_2$ 4 mM; CaCl$_2$ 1 mM; Tris 50 mM at pH 7.4; and enriched with cOmplete protease inhibitor) and treated with 8 U of microccocal nuclease (MNase) (New England Biolabs, Cat.

No.: M0247S) for 15 min at 37 °C and stopped by adding 10 µL 10 x sonication buffer (Tris 10 mM, EDTA 1 mM, 0.1% SDS at pH 8.0 and enriched with cOmplete protease inhibitor) followed by 10 min incubation on ice. Chromatin was then fragmented using a Covaris E220 sonicator (200 cycles, 75 pip W for 15 min). After centrifugation, the supernatant was collected, and the DNA concentration measured. For each immunoprecipitation, 5 µg chromatin were incubated with 4 µg anti-H3 histone antibody (Abcam, Cat. No.: ab1791), for 2 h at 4 °C, followed by the addition of 25 µL magnetic Protein G beads (Promega, Cat. No. G747A), pre-equilibrated in sonication buffer, and overnight rotation at 4 °C. Beads were sequentially washed for 5 min each at RT with rotation in 500 µL of sonication buffer, High-salt buffer (HEPES 50 mM, NaCl 500 mM, EDTA 1 mM, Triton-X-100 1%, Na-deoxycholate 0.1%, SDS 0.1% and PMSF0.5 mM at pH 7.9), Li buffer (Tris 20 mM, EDTA 1 mM, LiCl 250 mM, NP-40 0.5%, Na-deoxycholate 0.5%, PMSF 0.5 mM at pH 8.0], and twice with TE buffer (10 mM Tris and EDTA 1 mM at pH 8.0). DNA was eluted with 100 µL elution buffer (Tris 50 mM, 1 mM EDTA, SDS 1% SDS, NaCl 300 mM, NaHCO$_3$ 50 mM at pH 8.0) containing 2 µL Proteinase K (20 mg/mL) and incubated overnight at 65 °C to reverse crosslinking. DNA was finally purified the following day using 150 µl Ampure XP beads and eluted in 20 µL H$_2$O for quantification via 2200 TapeStation. Libraries were prepared using at least 25 ng of each sample via the Watchmaker kit (Watchmaker Genomics, Cat. No. 7K0102-024), with IDT xGen UDI-UMI Adapters (Integrated DNA Technologies, Cat. No. 10005925) and sequenced on an Illumina NovaSeq X + 10 B platform using a Paired-end 100 reads (PE100; Supplementary Table S4). Raw reads were aligned using bwa (v0.7.15) with default parameters and 12 computational threads (-t 12). The resulting SAM output was converted to BAM format, sorted and indexed using Samtools (v1.5) and transformed into BED files using bedtools (v2.24.0). From here, data were analyzed using the NucTools package following the published pipeline[93]. First, reads were sorted by name and the occupancy of the scaffolds containing genes was calculated using a window of 10 bp. Occupancy was calculated per scaffold, in an analogous way as the per chromosome way established by the pipeline generating .occ files and normalized per library size and scaffold length. Normalized occupancy levels for the first 1500 bp were extracted per gene and averaged per sample to produce aggregated profiles. Kolgomorov-Smirnov statistical tests were used to assess differences in the nucleosome distribution, and Wilcoxon Rank sum tests to evaluate differential occupation. Stability and fuzziness were computed for all the replicates of each treatment group using a variation of the stable_nucs_replicates.pl script including a modification in line 277 to capture only fuzzy regions whose relative error was higher than a determined threshold. Thresholds for stability were 0.2, and for, fuzziness, 1.5. The percentage of stable/fuzzy regions of each scaffold were calculated and compared using a Wilcoxon-Rank Sum test in R.

## Reporting summary

Further information on research design is available in the Nature Portfolio Reporting Summary linked to this article.

## Data availability

Single-cell RNA sequencing (scRNA-seq) data generated in this study have been deposited in the NCBI Gene Expression Omnibus (GEO) under accession code **GSE295870** [https://www.ncbi.nlm.nih.gov/geo/query/acc.cgi?acc=GSE295870]. Whole-genome bisulfite sequencing (WGBS) data generated in this study are available in GEO under accession code **GSE295869** [https://www.ncbi.nlm.nih.gov/geo/query/acc.cgi?acc=GSE295869]. Bulk RNA sequencing (RNA-seq) data generated in this study have been deposited in GEO under accession code **GSE295871** [https://www.ncbi.nlm.nih.gov/geo/query/acc.cgi?acc=GSE295871]. MNase-seq data generated in this study are available in GEO under accession code **GSE295867** [https://www.ncbi.nlm.nih.gov/geo/query/acc.cgi?acc=GSE295867]. The image cytometry

data generated in this study have been deposited in the Zenodo database under (https://doi.org/10.5281/zenodo.18771566). Source data are provided in this paper.

## Code availability

Data analyses were performed using publicly available software packages and published analysis pipelines as described in the Methods section. No custom software was developed for this study.

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

## Acknowledgements

We gratefully acknowledge Laura Wiegand, Areeba Khan and Nicole Schneider for technical support with the knockdown and behavioral experiments, as well as Steffen Schmitt and Tobias Rubner from the DKFZ Flow Cytometry Core Facility for support and guidance with the Image Cytometer. We also thank the Genomics and Proteomics Core Facility and the Single-Cell Open Lab for assistance with the sequencing experiments, as well as Jan-Phillip Mallm, Karsten Rippe and Vladimir Teif for the help with the MNase-Seq protocol and analysis using NucTools.

## Author contributions

J.J.D.L and V.C collected samples, performed experiments and analyzed the data. K.H. and G.R. provided technical and bioinformatical assistance, respectively. V.C. and F.L conceived the study. J.J.D.L and F.L wrote the paper with input from other authors. All authors approved the final manuscript.

## Funding

## Competing interests

The authors declare no competing interests.
