## [Transparent Peer Review file · Nature Communications]

Dnmt1 mediates epigenetic restriction of invasive traits in clonal crayfish

Corresponding Author: Professor Frank Lyko

Version 0:

Reviewer comments:

Reviewer #1

(Remarks to the Author)

The article is very interesting, clearly and fluently written. A wide variety of analyses have been performed to support the claims made. The methodology is well described, and the statistical analyses are appropriate. Overall, the entire article is very well developed, and only a few minor changes are required before final acceptance (see the attached PDF file for details on the minor corrections). One point I consider important is a final section listing the most frequently used abbreviations throughout the manuscript (those repeated more than twice). Otherwise, everything seems correct. Many congratulations to the authors for their excellent work.

Reviewer #2

(Remarks to the Author)

This manuscript examines how the Dnmt1 gene contributes to epigenetic regulation underlying invasiveness mechanisms in the marbled crayfish (*Procambarus virginalis*). This work also investigates the impact of environmental changes on Dnmt1 expression and their consequences for phenotypes.

To address this question, the authors combined complementary approaches, including RNAi-based knockdown, immune cell characterization by cytometry, single-cell RNA sequencing, and bisulfite whole-genome sequencing.

This is overall a very strong and well-executed study. The writing is clear and enjoyable to read. While the range of analytical approaches could appear overwhelming, the explanations are sufficiently detailed to ensure clarity. The introduction is particularly well structured, with a smooth flow that clearly highlights the underlying scientific problem.

The manuscript provides complete analytical information (sample sizes, statistical methods, significance criteria). Results are well substantiated, and the figures and tables (including supMat) are all correctly referenced in the text. The graphical presentation is clear and adequate, aside from a few minor typos that I detail below.

Line 61: ‘)’ maybe a typo.

Line 662 (figure 2 legend): “This assay was performed)” is a typo ? What the authors wanted to explain here ?

Legend figure 3. Although group colors for plots a,c,d,e and f seems to be the same as f color legend, maybe it would be possible to indicate this information within the legend in order to avoid any misleading (even if I agree that it’s intuitive).

Data availability => Please add in which platform raw data will be positioned (ncbi ? Other ?).

Reviewer #3

(Remarks to the Author)

• What are the noteworthy results?

Diaz-Larrosa et al. present a highly interesting and relevant study investigating how epigenetic regulation contributes to the invasiveness of marbled crayfish (*Procambarus virginalis*), a globally invasive, parthenogenetic species with a monoclonal genome. The study provides novel mechanistic insight by identifying Dnmt1-mediated DNA methylation as a critical regulator of phenotypic stability (“canalization”) in the marbled crayfish. Environmental downregulation or experimental

knockdown of Dnmt1 leads to global DNA hypomethylation, altered chromatin organization, and transcriptional deregulation of genes linked to immune and neural functions. Reduced methylation enhances behavioral traits linked to invasion success—boldness, activity, and exploration—demonstrating a connection between epigenetic relaxation and invasive potential. The findings show that Dnmt1 loss biases hemocyte differentiation toward granular immune cells while depleting neuronal precursors, resulting in a more robust immune system and altered neurogenesis. This bridges molecular, cellular, and behavioral layers of adaptation. The study supports the concept of the “General-Purpose Genotype”, proposing that epigenetic plasticity compensates for genetic uniformity in clonal species, enabling rapid responses to environmental challenges. By linking DNA methylation dynamics with ecological adaptability, the work provides a model for how epigenetic mechanisms facilitate biological invasions, offering a framework applicable to other clonal or low-diversity invaders.

- Will the work be of significance to the field and related fields? How does it compare to the established literature? If the work is not original, please provide relevant references.

The manuscript provides a significant contribution to the field of invasion biology, because it provides a mechanistic, epigenetic explanation for how clonal invasive species can achieve high adaptability and ecological success despite minimal genetic diversity. This research significantly advances invasion biology by shifting focus from purely genetic to epigenetic determinants of invasiveness, showing how DNA methylation plasticity drives behavioral, immunological, and ecological flexibility in a globally invasive species. However, the ecological components of this research could be outlined more in the introduction by mentioning the 2 central hypothesis addressed, namely the concept of the “General-Purpose Genotype” and the “Evolution of Increased Competitive Ability” model. See more specific comments below.

- Does the work support the conclusions and claims, or is additional evidence needed?

The manuscript presents a comprehensive and technically advanced study integrating behavioral assays with multi-omics (WGBS, RNA-seq, scRNA-seq, and MNase-seq) analyses to elucidate the epigenetic regulation of invasiveness in *P. virginalis*. The data analysis is methodologically sound, employing appropriate statistical frameworks and bioinformatic tools. However, several aspects of interpretation and conclusion require tempering. While the observed correlations between Dnmt1 knockdown, global hypomethylation, transcriptional dysregulation, and behavioral changes are compelling, the causal links among these levels remain inferred rather than experimentally demonstrated. Sample sizes in the omics datasets ($n = 2/3$ per group, depending on the method) limit statistical power, and some behavioral effects are modest and based on small cohorts. For the nucleosome analysis, the sample size is not stated and should be added. The extension of these results to general principles of invasiveness and immune advantage should be framed more cautiously, as ecological fitness was not directly assessed. Overall, the study is scientifically rigorous and provides a pioneering epigenetic framework for invasion biology, but its broader mechanistic and ecological conclusions should be presented as hypotheses for future validation rather than definitive claims. See more detailed comments below.

- Are there any flaws in the data analysis, interpretation and conclusions? Do these prohibit publication or require revision?

The data analyses in the manuscript are largely robust, but not without a few weaknesses or overextensions (see below), requiring revisions, mainly in terms of rewording, prior publication. The data interpretation is mainly biologically coherent: environmental Dnmt1 downregulation → hypomethylation → chromatin destabilization → altered differentiation → behavioral plasticity. The research elegantly connects molecular and ecological scales — a rare achievement in invasion biology. However, there are in our opinion a few overinterpretations that should be toned down. Below we give some reasons why/where we think that the presented results should be in parts interpreted more cautiously:

- Correlation vs. causation in methylation–expression relationships: The study reports that gene body hypomethylation correlates with both up- and downregulation of genes but interprets this as a causal regulatory effect of Dnmt1. However, correlation between methylation and transcription is not necessarily causative, especially when Dnmt1 knockdown may affect transcription through indirect chromatin remodeling or RNAi off-target effects. Without rescue experiments (e.g., Dnmt1 re-expression) or temporal sequencing of methylation vs. transcriptional changes, causality remains suggestive, not proven.
- Sample sizes in multi-omics datasets: WGBS, RNA-seq, and MNase-seq each use only low numbers of biological replicates per group, which is below standard statistical power for genome-wide analysis. Although replication is high at the cellular and molecular level, the low n increases the risk of false positives in identifying differentially methylated or expressed genes. These number of replicates is also not very clearly reported, and we recommend to include an overview table of sample replicates per experiment/method.
- scRNA-seq integration and cell-type annotation: Cell-type identities are inferred via homology to genes from other crustaceans (e.g., *Pacifastacus leniusculus*), but functional validation (e.g., FACS, immunostaining) is missing. This is acceptable for exploratory work but weakens the certainty that “hemocyte-derived neuronal precursors” (HDNPs) are truly neurogenic cells rather than transcriptionally similar subtypes.
- Behavioral analysis statistics: Behavioral parameters are normalized and compared across relatively small groups ($n = 8–9$). Some parameters show only marginal p -values (e.g., $p = 0.06$ for “stop average duration”). Yet, these are still interpreted as meaningful behavioral enhancement. The study could have benefited from multivariate behavioral models or power analysis to substantiate these effects.
- MNase-seq interpretation: The claim that “nucleosome destabilization underlies transcriptional changes” is plausible but not directly demonstrated — correlation of fuzziness with expression change is indirect evidence.

- “Epigenetic canalization” claim: The concept that Dnmt1 maintains phenotypic canalization is intriguing but speculative. Demonstrating true canalization requires developmental trajectory experiments or quantitative measures of phenotypic variance, not just observation of altered differentiation after knockdown. Also, the introduction would benefit from mentioning

such general hypothesis from the start.

- Behavioral causality: Enhanced boldness and activity are attributed to reduced Dnmt1 and downstream neurogenic effects. However, no direct neurophysiological measurements (e.g., neuronal firing, neurotransmitter assays) support this link. The behavioral phenotype could result from systemic metabolic stress or immune activation rather than neurogenesis impairment.

- Ecological generalization: The conclusion that “epigenetic relaxation promotes invasiveness” is reasonable but extrapolates from one clonal species to invasiveness in general. This generalization would need comparative or field-based validation (e.g., across multiple invasive crayfish or other clonal taxa).

- EICA hypothesis alignment: The discussion links the immune findings to the “Evolution of Increased Competitive Ability” model. However, whether increased GC counts truly improve fitness or competitive ability was not tested. The immune enhancement is inferred, not demonstrated. Again, the introduction would benefit for the reader to have a mentioning of the EICA hypothesis.

• Is the methodology sound? Does the work meet the expected standards in your field?

The methodology described is sound, rigorous, state-of-the-art and cutting-edge for non-model invertebrates. RNAi, adapted to crustaceans using optimized injection routes and validated via tissue-wide qPCR, represents a rare but solid functional genomics tool in non-model invertebrates. High-coverage, genome-wide methylation profiling at single-base resolution is the standard for methylome analysis. scRNA-seq using 10x Genomics platform is a cutting-edge method for cell-type resolution, enabling the identification of hemocyte subpopulations and differentiation trajectories. MNase-seq used to probe nucleosome positioning is an advanced approach that links chromatin architecture to transcriptional outcomes. Especially the integration of WGBS + RNA-seq + MNase-seq is convincingly combining epigenomic and transcriptomic layers, analytically sophisticated and current in systems epigenetics.

As minor caveats we want to mention, however, that the reference genome annotation for *P. virginalis* remains incomplete, which can limit functional interpretation of omics data. The authors mitigate this with cross-species BLAST annotation, a reasonable workaround but not ideal for pathway analysis.

• Is there enough detail provided in the methods for the work to be reproduced?

The methods are detailed and include many essential reagents, instruments, kit names, parameter values, software versions and QC steps, but a few missing specifics should be added to make full, unambiguous reproduction straightforward.

- Full bioinformatics parameter sets and scripts: please provide exact mapping parameters used for WGBS (e.g., aligner and arguments), bisulfite mapping tool/version, DMR calling method and thresholds (they mention $|\Delta\text{methylation}| > 0.1$ but not statistical cutoffs), read filtering thresholds, and command-line options for Hisat2/CellRanger/NucTools. Provide the analysis scripts (R, python, shell) or a GitHub repository and mention random seeds used for any stochastic steps (important e.g., for PCA/UMAP/seurat integration).

- Exact single-cell filtering thresholds & post-filter cell counts: the authors describe filters (mitochondrial $>5\%$, >2000 or <20 features), but reproducibility needs the final per-sample cell counts after filtering, and the exact Seurat parameters used for normalization, variable feature selection (they gave top 2,000), dimensionality used, and clustering resolution(s).

- MNase digestion and sonication specifics: the authors state “8 U MNase” and use Covaris E220 but do not give MNase digestion time/temperature, Covaris settings (duty cycle, cycles per burst, time), crosslinking conditions (concentration/time), or how samples were normalized between libraries. Please add exact experimental settings.

- WGBS library prep and mapping QC thresholds: State bisulfite conversion QC metrics (unconversion rate threshold), minimal coverage cutoffs (the authors mention filtering out cytosines with $<10X$ combined coverage — good, but confirm how combined across replicates was treated), adapter-trimming settings and mapping software for bisulfite reads (and versions).

- Exact behavioral sample sizes and exclusion criteria: The authors note “animals with no locomotion were not analyzed” — state how many were excluded and final N per group per batch in a table. Also provide any blinding/randomization procedures used during behavioral scoring.

- QC metrics for sequencing libraries (e.g., insert size distributions, average depth per sample, mapping rates) is usually included as Supplementary Tables.

- Power analysis or rationale for chosen sample sizes for behavioral assays and omics replicates (acknowledging constraints for non-model organisms).

- Ethical permit ID numbers (they say approved; please include approval reference).

Additional specific comments:

L46-47: This sentence is not reflecting well the content of this paragraph. We suggest to use this sentence for starting a specific additional paragraph on behavioural traits in invasiveness, and include here also the EICA hypothesis and the General-Purpose Genotype hypothesis.

L60: delete closing bracket „)“

L64: replace „sibling species“ with „parent species“ to be consistent with the discussion wording.

L79: For crayfish you stated first the scientific name, than the common name in brackets - stay consistent.

L83ff: the introduction would clearly benefit from the statement of working hypotheses to be tested.

L98: It is not clear from reading the introduction why the authors have done a temperature experiment. The ecological reasoning is missing from the intro and the discussion. It is only mentioned in the method section (LL396-403).

L106: It is also not clear from the introduction why a biofloc environment was tested. What was the stressor here? Please clarify.

L107: was the Dnmt1 expression in hemocytes done with qPCR? qPCR is mentioned in the next subsection as method, but not here, please clarify.

L117: via „in“ vitro

L127: replace knockdown with „KD“ as it is introduced.

L177: no space between 55 and % (general comment, please check throughout the manuscript)

L232: The other KD samples were taken 28 dpi. And here they were taken 1 and 6 months. This is not mentioned in the methods. What is the reason for the 6 months?

LL245-247: In the methods is mentioned supplement material S6, where a lot of other genes are listed. Why discussing only these genes presented here, while the other results are only mentioned in supplement figure S2d?

L258: There is no figure s2f.

L329: replace knockdown with „KD“

L349: replace knockdown with „KD“

L351: do not start a sentence with abbreviated genus name.

L364: please specify whether the non-invasive population was from the same species

LL376-378: can you provide any hypothesis for the potential impact on adult neurogenesis?

L379: this is a very blunt final statement. Please specify how this research could benefit strategies for invasion management, as this seems a bit far fetched.

Best regards,

Reviewer #4

(Remarks to the Author)

Version 1:

Reviewer comments:

Reviewer #3

(Remarks to the Author)

We have reviewed the revised manuscript and the authors' response letter. We are satisfied with the way the authors have addressed our previous comments. The additions regarding the methodology and study design, as well as the explanations in the introduction, have clearly improved the clarity of the manuscript.

Regarding methodology, the sample size details are now included in the supplement, the parameters of bioinformatic analyses have been added and explained in detail. Statements regarding causality of invasive traits and epigenetic modification have been toned down. Important contextual information about epigenetic canalisation, EICA and GPG hypothesis was added in the introduction.

We recommend acceptance at this stage, assuming the minor changes below are made:

1. We appreciate the clarification that the bioinformatic analyses were repeated. To strengthen the robustness, please specify how many times the analyses using random clustering were repeated.
2. The addition of the hypothesis addresses our comments. Please consider expanding this section to clearly state the ecological parameters tested: temperature and water quality (different composition in nutrients and microbial exposure)

Reviewer #4

(Remarks to the Author)

Response to reviewers' comments

Reviewer #1

The article is very interesting, clearly and fluently written. A wide variety of analyses have been performed to support the claims made. The methodology is well described, and the statistical analyses are appropriate. Overall, the entire article is very well developed, and only a few minor changes are required before final acceptance (see the attached PDF file for details on the minor corrections). One point I consider important is a final section listing the most frequently used abbreviations throughout the manuscript (those repeated more than twice). Otherwise, everything seems correct. Many congratulations to the authors for their excellent work.

>> We thank the reviewer for these very encouraging comments. We accepted the large majority of the suggested edits into the manuscript. A comprehensive list of abbreviations is now provided as an appendix at the end of the Supplementary Information document.

Reviewer #2

This manuscript examines how the Dnmt1 gene contributes to epigenetic regulation underlying invasiveness mechanisms in the marbled crayfish (*Procambarus virginalis*). This work also investigates the impact of environmental changes on Dnmt1 expression and their consequences for phenotypes.

To address this question, the authors combined complementary approaches, including RNAi-based knockdown, immune cell characterization by cytometry, single-cell RNA sequencing, and bisulfite whole-genome sequencing.

This is overall a very strong and well-executed study. The writing is clear and enjoyable to read. While the range of analytical approaches could appear overwhelming, the explanations are sufficiently detailed to ensure clarity. The introduction is particularly well structured, with a smooth flow that clearly highlights the underlying scientific problem.

The manuscript provides complete analytical information (sample sizes, statistical methods, significance criteria). Results are well substantiated, and the figures and tables (including supMat) are all correctly referenced in the text. The graphical presentation is clear and adequate, aside from a few minor typos that I detail below.

>> We thank the reviewer for these very encouraging comments.

Line 61: ')' maybe a typo.

>> The extra parenthesis has been removed.

Line 662 (figure 2 legend): “This assay was performed).” is a typo? What the authors wanted to explain here?

>> The incomplete phrase has been removed.

Legend figure 3. Although group colors for plots a,c,d,e and f seems to be the same as f color legend, maybe it would be possible to indicate this information within the legend in order to avoid any misleading (even if I agree that it's intuitive).

>> Clarified as suggested.

Data availability => Please add in which platform raw data will be positioned (ncbi? Other?).

>> Specified as requested in LL674-675: “scRNA-seq, WGBS, RNA-seq and MNase-seq data are available from the NCBI Gene Expression Omnibus (GEO).”.

Reviewer #3

We thank the reviewer for her very encouraging general comments. A detailed point-by-point response to the specific comments is provided below:

1. Correlation vs. causation in methylation–expression relationships: The study reports that gene body hypomethylation correlates with both up- and downregulation of genes but interprets this as a causal regulatory effect of Dnmt1. However, correlation between methylation and transcription is not necessarily causative, especially when Dnmt1 knockdown may affect transcription through indirect chromatin remodeling or RNAi off-target effects. Without rescue experiments (e.g., Dnmt1 re-expression) or temporal sequencing of methylation vs. transcriptional changes, causality remains suggestive, not proven.

>> We have toned down the corresponding text to avoid implying direct cause-and-effect relationships. In particular, we replaced “are underpinned by” with “coincide with” in the *Abstract* (L19) and clarified in the *Discussion* (LL383-386) that a causal relationship would require further experimental validation, such as time-resolved analyses or Dnmt1 rescue experiments.

2. Sample sizes in multi-omics datasets: WGBS, RNA-seq, and MNase-seq each use only low numbers of biological replicates per group, which is below standard statistical power for genome-wide analysis. Although replication is high at the cellular and molecular level, the low

n increases the risk of false positives in identifying differentially methylated or expressed genes. These number of replicates is also not very clearly reported, and we recommend to include an overview table of sample replicates per experiment/method.

>> A table containing the number of biological replicates per experiment/method has now been included as suggested as Supplementary table S5. Additionally, the main text has been revised, and sample sizes have been explicitly stated where appropriate: *Results L234; LL254-255*.

3. scRNA-seq integration and cell-type annotation: Cell-type identities are inferred via homology to genes from other crustaceans (e.g., *Pacifastacus leniusculus*), but functional validation (e.g., FACS, immunostaining) is missing. This is acceptable for exploratory work but weakens the certainty that “hemocyte-derived neuronal precursors” (HDNPs) are truly neurogenic cells rather than transcriptionally similar subtypes.

>> We have now clarified this point in the *Results (LL198-199)* and *Methods (LL542-545)* sections. Specifically, we state that cluster annotations were inferred using differentially expressed gene sets and homology-based comparisons to published crustacean datasets. In the discussion, we emphasize that these identities should be considered *putative*.

4. Behavioral analysis statistics: Behavioral parameters are normalized and compared across relatively small groups (n = 8–9). Some parameters show only marginal p-values (e.g., p = 0.06 for “stop average duration”). Yet, these are still interpreted as meaningful behavioral enhancement. The study could have benefited from multivariate behavioral models or power analysis to substantiate these effects.

>> We would like to clarify that a multivariate behavioral analysis was already performed in the original version of the manuscript, as described by Fossat *et al.* (2014). The results revealed a statistically significant separation between control and *Dnmt1* KD groups (p<0.05; Monte-Carlo permutation test) supporting a global difference in behavior.

5. MNase-seq interpretation: The claim that “nucleosome destabilization underlies transcriptional changes” is plausible but not directly demonstrated — correlation of fuzziness with expression change is indirect evidence.

>> We have revised the text to describe nucleosome destabilization as correlating with transcriptional changes in the *Results* section (LL310-312) and added a statement in the *Discussion (LL383-386)* noting that direct mechanistic evidence would require further experiments. Additionally, we replaced “key mechanism influencing transcriptional changes” with “mechanism associated with transcriptional changes” in the *Abstract (LL20-21)*.

6. “Epigenetic canalization” claim: The concept that Dnmt1 maintains phenotypic canalization is intriguing but speculative. Demonstrating true canalization requires developmental trajectory experiments or quantitative measures of phenotypic variance, not just observation of altered differentiation after knockdown. Also, the introduction would benefit from mentioning such general hypothesis from the start.

>> We have now added a brief explanation of the canalization concept to the *Introduction* (LL60-62) and toned down the interpretation in the *Discussion*.

7. Behavioral causality: Enhanced boldness and activity are attributed to reduced Dnmt1 and downstream neurogenic effects. However, no direct neurophysiological measurements (e.g., neuronal firing, neurotransmitter assays) support this link. The behavioral phenotype could result from systemic metabolic stress or immune activation rather than neurogenesis impairment.

>> Several observations support that the behavioral phenotype is *Dnmt1*-dependent rather than a nonspecific response. First, the dsGFP control accounts for handling and general dsRNA-related effects. Second, no differences were observed in survival, weight, reproduction, or general health, and transcriptomic data did not indicate broad metabolic changes. Finally, systemic immune activation typically reduces locomotion and exploratory behavior (Gruber et al., 2014)¹, contrasting with the increased activity observed here. Nevertheless, we have now used more cautious wording in the *Discussion* (L333; “underpinned by” replaced with “accompanied”).

1. Gruber, C. et al. Variation in resistance to the invasive crayfish plague and immune defence in the native noble crayfish. *Ann. Zool. Fenn.* **51**, 371–389 (2014).

8. Ecological generalization: The conclusion that “epigenetic relaxation promotes invasiveness” is reasonable but extrapolates from one clonal species to invasiveness in general. This generalization would need comparative or field-based validation (e.g., across multiple invasive crayfish or other clonal taxa).

>> We have now restricted the sentence to *P. virginalis* (L372).

9. EICA hypothesis alignment: The discussion links the immune findings to the “Evolution of Increased Competitive Ability” model. However, whether increased GC counts truly improve fitness or competitive ability was not tested. The immune enhancement is inferred, not demonstrated. Again, the introduction would benefit for the reader to have a mentioning of the EICA hypothesis.

>> We agree that our data only allows inference of a potentially stronger immune system in *Dnmt1* KD animals. Accordingly, we now phrase this as a suggestion in the *Discussion* (LL360-362), saying the increase of GC counts and the upregulation of effector genes in our *Dnmt1* KD crayfish also suggest a *potentially* stronger immune system, which could be advantageous in new environments. To improve clarity, we now also introduce the *Evolution of Increased Competitive Ability (EICA)* hypothesis earlier in the manuscript, in the context of physiological adaptability (*Introduction*, LL53–56).

10. As minor caveats we want to mention, however, that the reference genome annotation for *P. virginalis* remains incomplete, which can limit functional interpretation of omics data. The authors mitigate this with cross-species BLAST annotation, a reasonable workaround but not ideal for pathway analysis.

>> We agree with the reasoning of the reviewer and have provided a detailed description of our approach in the *Methods* section.

11. Full bioinformatics parameter sets and scripts: please provide exact mapping parameters used for WGBS (e.g., aligner and arguments), bisulfite mapping tool/version, DMR calling method and thresholds (they mention $|\Delta\text{methylation}| > 0.1$ but not statistical cutoffs), read filtering thresholds, and command-line options for Hisat2/CellRanger/NucTools. Provide the analysis scripts (R, python, shell) or a GitHub repository and mention random seeds used for any stochastic steps (important e.g., for PCA/UMAP/seurat integration).

>> The requested bioinformatic parameters have been included under each specific *Methods* subsection: scRNA-seq (LL527-529); WGBS/MiSeq (LL565-572; 593-594); RNA-seq (LL609-610; NucTools (LL648-650). The different scripts from NucTools can be found in the mentioned reference or in <https://homeveg.github.io/nuctools/>. Pipeline and scripts were ran as established with the changes, thresholds and parameters mentioned in the *Methods* section. No manual random seed was set, but all analyses were repeated to confirm result stability.

12. Exact single-cell filtering thresholds & post-filter cell counts: the authors describe filters (mitochondrial >5%, >2000 or <20 features), but reproducibility needs the final per-sample cell counts after filtering, and the exact Seurat parameters used for normalization, variable feature selection (they gave top 2,000), dimensionality used, and clustering resolution(s).

>> Pre- and post-filter cell counts for each sample are reported in Supplementary Table S1. In addition, we have expanded the *Methods* section (LL536-539) to include the exact Seurat parameters used for normalization, feature selection, and clustering. For the integration of the 3 control and 3 KD samples, we also added additional information to the *Methods* (L533):

“We downsampled each replicate to 622 cells, the number of cells of the least represented sample”.

13. MNase digestion and sonication specifics: the authors state “8 U MNase” and use Covaris E220 but do not give MNase digestion time/temperature, Covaris settings (duty cycle, cycles per burst, time), crosslinking conditions (concentration/time), or how samples were normalized between libraries. Please add exact experimental settings.

>> Detailed experimental parameters have been added to the *Methods* (LL622-645), as requested.

14. WGBS library prep and mapping QC thresholds: State bisulfite conversion QC metrics (unconversion rate threshold), minimal coverage cutoffs (the authors mention filtering out cytosines with <10X combined coverage — good, but confirm how combined across replicates was treated), adapter-trimming settings and mapping software for bisulfite reads (and versions).

>> All relevant QC metrics, including raw read numbers, mapping efficiency, sequencing depth, post-filtering coverage, and bisulfite conversion ratios, are provided in Supplementary Table S2. Bisulfite conversion efficiency exceeded 99.7 % for all samples. As stated in the *Methods*, cytosines with a coverage below 10X were excluded from further analysis, and only sites covered in all samples were kept. Adapter trimming settings and mapping software with versions have now been included in the *Methods* section (LL565-572).

15. Exact behavioral sample sizes and exclusion criteria: The authors note “animals with no locomotion were not analyzed” — state how many were excluded and final N per group per batch in a table. Also provide any blinding/randomization procedures used during behavioral scoring.

>> Three independent batches of animals with similar body weight were recorded and analyzed (three control and three KD animals per batch; total n=9 per group). One KD animal exhibited no locomotion and was therefore excluded from analysis, following the same exclusion criterion as other studies (Fossat et al., 2014). Animal identities were revealed only after completion of data analysis for treatment group assignment and comparison. This information has now been added to the *Methods* (LL476; LL487-489).

16. QC metrics for sequencing libraries (e.g., insert size distributions, average depth per sample, mapping rates) is usually included as Supplementary Tables.

>> Comprehensive QC metrics have been provided in the Supplementary Information. Specifically, Supplementary Tables S1-S4 include QC statistics for the scRNA-seq, WGBS, RNA-seq, and MNase-seq libraries, respectively.

17. Power analysis or rationale for chosen sample sizes for behavioral assays and omics replicates (acknowledging constraints for non-model organisms).

>> Sample sizes were determined by feasibility constraints inherent to behavioral and omics work in non-model organisms. For behavioral assays, 18 animals (9 per group) were analyzed across 3 independent rounds of recordings, limited by the availability of 6 isolation tanks. This design was sufficient to detect consistent behavioral trends. For omics datasets, we used 3 control and 3 KD biological replicates for scRNA-seq, 4 control and 6 KD samples for MNase-seq, and a matched set of 2 control and 2 KD samples for WGBS and RNA-seq, respectively to enable direct per-sample comparisons. Differential methylation and gene expression was extensively validated by amplicon sequencing and qPCR, respectively, in 3 or more independent samples.

18. Ethical permit ID numbers (they say approved; please include approval reference).

>> Ethical permit numbers have now been provided (*Methods; L410*).

19. L46-47: This sentence is not reflecting well the content of this paragraph. We suggest to use this sentence for starting a specific additional paragraph on behavioural traits in invasiveness, and include here also the EICA hypothesis and the General-Purpose Genotype hypothesis.

>> We have now restructured the *Introduction (LL53-62)* to create a distinct paragraph focusing on behavioural and physiological flexibility as key features of invasive success. We use this to then introduce the *Increased Competitive Ability (EICA)* and *General-Purpose Genotype (GPG)* hypotheses providing a connection between phenotypic plasticity, ecology and DNA methylation as a mediator of advantageous processes in invasiveness.

20. L60: delete closing bracket „)“

>> Corrected as suggested.

21. L64: replace „sibling species“ with „parent species“ to be consistent with the discussion wording.

>> Corrected as suggested.

22. L79: For crayfish you stated first the scientific name, than the common name in brackets - stay consistent.

>> Corrected as suggested, throughout the manuscript.

23. L83ff: the introduction would clearly benefit from the statement of working hypotheses to be tested.

>> A clear statement of the working hypothesis has been included at the end of the *Introduction* (LL92-93).

24. L98: It is not clear from reading the introduction why the authors have done a temperature experiment. The ecological reasoning is missing from the intro and the discussion. It is only mentioned in the *Methods* section (LL396-403).

>> We used temperature fluctuations to test whether *Dnmt1* expression responds to rapid environmental changes, which can be encountered by invasive species during dispersal. Temperature is a biologically relevant and easily controlled parameter, which has already been shown to affect crayfish methylome (Tönges et al., 2021). We have now added a short explanation in the *Introduction* (LL93-95) stating this rationale.

25. L106: It is also not clear from the introduction why a biofloc environment was tested. What was the stressor here? Please clarify.

>> Biofloc systems are semiautonomous environments containing diverse microbial and algal communities that can influence crayfish methylome (Coutinho *et al.*, 2025). Transitioning animals from a biofloc to a clearwater aquarium environment, thus mimics an ecologically relevant shift in water quality, nutrient composition, and microbial exposure. We have now clarified this rationale in the *Introduction* (LL93-95).

26. L107: was the *Dnmt1* expression in hemocytes done with qPCR? qPCR is mentioned in the next subsection as method, but not here, please clarify.

>> Clarified as suggested.

27. L117: via „in“ vitro

>> Corrected as suggested.

28. L127: replace knockdown with „KD“ as it is introduced.

>> Corrected as suggested.

29. L177: no space between 55 and % (general comment, please check throughout the manuscript)

>> Corrected as suggested, throughout the manuscript.

30. L232: The other KD samples were taken 28 dpi. And here they were taken 1 and 6 months. This is not mentioned in the methods. What is the reason for the 6 months?

>> "1 month" corresponds to 28 days post-injection. The 6-month time point was included to assess the persistence of methylation changes following Dnmt1 KD. This has now been clarified in the *Results*. L247.

31. LL245-247: In the methods is mentioned supplement material S6, where a lot of other genes are listed. Why discussing only these genes presented here, while the other results are only mentioned in supplement figure S2d?

>> The highlighted genes from Figure 4e (ProPO, SOD, Notch, klumpfuss) were selected for discussion because they are directly involved in arthropod hematopoiesis and immunity, while the rest of the validated genes have other functions. This has now been clarified in the *Results* (L260).

32. L258: There is no figure s2f.

>> Two panels were mistakenly labeled as S2d and there was no S2f in the Supplementary Figure. This has now been corrected.

33. L329: replace knockdown with „KD“

>> Corrected as suggested.

34. L349: replace knockdown with „KD“

>> Corrected as suggested.

35. L351: do not start a sentence with abbreviated genus name.

>> Corrected as suggested.

36. L364: please specify whether the non-invasive population was from the same species

>> The comparisons discussed refer to populations of the same species. For the tubeworm and mussel, the comparison is made between newly and long-established populations, while in the case of the ascidian, it compares invasive to non-invasive populations. We have clarified this in the *Discussion* (LL374-377).

37. LL376-378: can you provide any hypothesis for the potential impact on adult neurogenesis?

>> We have clarified this point in the *Discussion* (LL394-398). We suggest that a decrease in neuronal renewal, which mainly targets clusters innervating the olfactory and accessory lobes that are essential for detecting and integrating olfactory, visual and mechanosensory signals (Beltz and Benton, 2017). This may alter sensory processing, stress responsiveness and overall behaviour, impacting adaptation to novel environments.

38. L379: this is a very blunt final statement. Please specify how this research could benefit strategies for invasion management, as this seems a bit far fetched.

>> We have rewritten the final paragraph.

Reviewer #4

>> We thank the reviewer for his/her/their support.

Second round of revisions

Reviewer #3

We thank the reviewer for her very encouraging general comments. A detailed point-by-point response to the specific comments is provided below:

1. We appreciate the clarification that the bioinformatic analyses were repeated. To strengthen the robustness, please specify how many times the analyses using random clustering were repeated.

>> The bioinformatic analyses involving stochastic steps (including dimensionality reduction and clustering in Seurat) were repeated multiple times using identical input data and parameters and the resulting clustering structure was consistent across runs. No explicit random seed was set during these analyses. This information has now been clarified in the *Methods* section under *scRNA-seq data analysis* subsection.

2. The addition of the hypothesis addresses our comments. Please consider expanding this section to clearly state the ecological parameters tested: temperature and water quality (different composition in nutrients and microbial exposure)

>> Ecological parameters tested are now stated at the end of the *Introduction* to increase clarity.